# Remodeling of Biomembranes and Vesicles by Adhesion of Condensate Droplets

**DOI:** 10.3390/membranes13020223

**Published:** 2023-02-10

**Authors:** Reinhard Lipowsky

**Affiliations:** Max Planck Institute of Colloids and Interfaces, Science Park Golm, 14424 Potsdam, Germany; lipowsky@mpikg.mpg.de

**Keywords:** synthetic biosystems, biomembranes, condensate droplets, adhesion, surface tensions, wetting transitions, membrane tubulation, engulfment, line tension, endocytosis

## Abstract

Condensate droplets are formed in aqueous solutions of macromolecules that undergo phase separation into two liquid phases. A well-studied example are solutions of the two polymers PEG and dextran which have been used for a long time in biochemical analysis and biotechnology. More recently, phase separation has also been observed in living cells where it leads to membrane-less or droplet-like organelles. In the latter case, the condensate droplets are enriched in certain types of proteins. Generic features of condensate droplets can be studied in simple binary mixtures, using molecular dynamics simulations. In this review, I address the interactions of condensate droplets with biomimetic and biological membranes. When a condensate droplet adheres to such a membrane, the membrane forms a contact line with the droplet and acquires a very high curvature close to this line. The contact angles along the contact line can be observed via light microscopy, lead to a classification of the possible adhesion morphologies, and determine the affinity contrast between the two coexisting liquid phases and the membrane. The remodeling processes generated by condensate droplets include wetting transitions, formation of membrane nanotubes as well as complete engulfment and endocytosis of the droplets by the membranes.

## 1. Introduction

The cells of our body are divided up into separate compartments by biomembranes that form closed surfaces and vesicles. The biomembranes represent molecular bilayers, which are fluid and have a thickness of a few nanometers. Even though these membranes provide robust barriers for the exchange of molecules between different compartments, they are highly flexible and can easily remodel their shape and topology. These remodeling processes can be systematically and quantitatively studied using biomimetic model systems [1,2,3]. In this review, we consider remodeling processes which are induced by the adhesion of condensate droplets arising from phase separation in aqueous solutions. The term ‘condensate droplet’ is used to emphasize that the droplet is bounded by a liquid–liquid rather than by a liquid–gas interface.

Liquid droplets adhering to solid or rigid substrates have been studied for a long time. Each droplet forms a certain contact angle with the substrate as described by Young’s equation, which was obtained more than 200 years ago [4]. For a solid or rigid substrate, one can ignore the elastic deformations of this substrate arising from the interactions with the droplet. In contrast, condensate droplets adhering to a biomembrane lead to strong elastic deformations of this membrane. The membrane forms a contact line with the droplet and acquires a very high curvature close to this line [5]. The vesicle-droplet system attains a variety of different adhesion morphologies and undergoes wetting transitions between these morphologies as we change the molecular composition or the temperature [6]. Particularly fascinating remodeling processes of membranes interacting with condensate droplets are the formation of membrane nanotubes [7,8,9], the formation of two daughter vesicles that enclose two different condensate droplets [10], and the complete engulfment of the droplets by the membranes [11].

Wetting transitions of condensate droplets at biomembranes were first observed when giant unilamellar vesicles (GUVs) were exposed to aqueous PEG-dextran solutions that separated into a PEG-rich and a dextran-rich phase [5,6,10,11]. Aqueous two-phase (or biphasic) systems based on biopolymers such as PEG and dextran have been applied for several decades in biochemical analysis and biotechnology [12] and are intimately related to water-in-water emulsions [13]. Aqueous phase separation within GUVs was first reported by Christine Keating and coworkers [14].

The PEG-dextran solutions undergo phase separation when the weight fractions of the polymers exceed a few percent. The corresponding interfacial tensions are very low, of the order of $10^{-4}$ to $10^{-1}$ mN/m, reflecting the vicinity of a critical demixing point in the phase diagram [15,16,17,18]. The aqueous phase separation of PEG-dextran solutions provides an example for *segregative* phase separation, in which one phase is enriched in one macromolecular component such as PEG whereas the other phase is enriched in the other macromolecular component such as dextran. The segregative behavior implies that the different species of macromolecules effectively repel each other. Another type of aqueous two-phase system is created by *associative* phase separation, for which one phase is enriched in the macromolecular components whereas the other phase represents a dilute aqueous solution of the macromolecules [19,20,21,22]. The associative behavior implies that the different macromolecular species effectively attract each other. Associative phase separation is observed, for instance, in solutions of two, oppositely charged polyelectrolytes [21,22], a process also known as coacervation, which leads to coacervate droplets enriched in the polyelectrolytes. Recently, the interactions of coacervate droplets with GUV membranes have also been studied. These studies include the formation of coacervate droplets within GUVs [23,24], the exocytosis of such droplets from GUVs [25,26], and the endocytosis and uptake of coacervate droplets by GUVs [27].

In this review, the framework of fluid elasticity is used to understand the mutual remodeling of biomembranes and condensate droplets. This framework is appealing from a conceptual point of view because it involves only two basic assumptions. The first assumption is that the condensate droplets are bounded by a liquid–liquid interface, arising from liquid–liquid phase separation. A liquid–liquid interface between the droplet and the second aqueous phase can be characterized by its interfacial tension, irrespective of whether the liquid droplet is formed by segregative or associative phase separation. The second assumption is that the biomembranes are in a fluid state which implies that their morphology is governed by a few curvature-elastic parameters such as their bending rigidity (or bending resistance) and their spontaneous (or preferred) curvature. When these two basic assumptions are fulfilled, the framework of fluid elasticity applies to the vesicle-droplet system irrespective of its molecular composition and irrespective of the underlying intermolecular interactions. Therefore, for the purpose of this review, coacervate droplets, which typically involve screened electrostatic interactions between oppositely charged macromolecules, will be considered as a special kind of condensate droplets.

Using the framework of fluid elasticity, one can identify the key parameters that determine the remodeling behavior of vesicle-droplet systems and obtain important relationships between these key parameters and the properties of these systems as measured in experimental studies and observed in computer simulations. The numerical values of the fluid-elastic parameters can then be deduced by combining these relationships with the results of the experiments and simulations.

During the last two decades, we have introduced and continuously developed the framework of fluid elasticity. Our studies were based on the combination of analytical theory, experimental observations, and computer simulations, reflecting my credo that real understanding requires the fruitful interplay of these different methods. As a result, we obtained an integrated view and identified the key parameters for the remodeling processes. In addition to the interfacial tension of the droplet and the curvature-elastic parameters of the membrane, we need to take the adhesion free energies between the two aqueous phases and the membrane into account as well as the line tension of the contact line [28]. The contact line of a vesicle-droplet system represents the narrow membrane segment in contact with the liquid–liquid interface, which exerts capillary forces onto this line. The associated line tension can be positive or negative as revealed by molecular dynamics simulations [29,30]. Furthermore, the sign of the line tension determines the shape of narrow or closed membrane necks that are formed during the exocytosis or endocytosis of condensate droplets.

Condensate droplets have also been observed in living cells where they provide separate liquid compartments which are not bounded by membranes. Examples for these condensates include germ P-bodies [31,32], nucleoli [33], and stress granules [34]. These biomolecular condensates are believed to form via liquid–liquid phase separation in the cytoplasm [31,35] and can be reconstituted in vitro [36,37,38,39]. They are enriched in certain types of proteins that have intrinsically disordered domains and interact via multivalent macromolecular interactions [35,38,39,40,41]. Remodeling of cellular membranes by condensate-membrane interactions has been observed for P-bodies that adhere to the outer nuclear membrane [31], for lipid vesicles within a synapsin-rich liquid phase [42], for TIS granules interacting with the endoplasmic reticulum [43], for condensates at the plasma membrane [44,45,46], and for condensates that are enriched in the RNA-binding protein Whi3 and adhere to the endoplasmatic reticulum [47].

Our discussion of condensate droplets in contact with biomembranes and vesicles starts with the geometry of these systems which involves three liquid phases α, β, and γ as shown in Figure 1. The two phases α and β are formed by segregative or associative liquid–liquid phase separation and are separated by the αβ interface. When the droplet adheres to the membrane, the αβ interface forms a contact line with the membrane, which divides the membrane up into two segments, the αγ segment exposed to the α and γ phases as well as the βγ segment in contact with the β and γ phases. In Figure 1a,b, the coexisting phases α and β are located outside and inside the vesicle, respectively.

To describe the vesicle-droplet morphology in a quantitative manner, we introduce three apparent contact angles that can be directly measured by (conventional) optical microscopy. These contact angles are intimately related to three surface tensions, $\Sigma_{\alpha \beta }$, $\Sigma_{\alpha \gamma }^{m}$, and $\Sigma_{\beta \gamma }^{m}$, which balance along the contact line and define the affinity contrast between the two coexisting liquid phases α and β in contact with the membrane. Even though the affinity contrast is a mechanical quantity, it can be obtained from the apparent contact angles, which represent purely geometric quantities. When we take the bending rigidity of the membrane into account, the three apparent contact angles are reduced to two intrinsic contact angles. The affinity contrast, which is still defined in terms of the three surface tensions, can now be expressed in terms of the intrinsic contact angles. The affinity contrast is shown to govern the onset of adhesion and to provide a global view of the different adhesion morphologies and wetting transitions. The interplay between the interfacial tension $\Sigma_{\alpha \beta }$, which pulls on the membrane via capillary forces, and the membrane’s bending rigidity κ, which acts to flatten the membrane, determine the high membrane curvature observed along the contact line, which is of the order of $\sqrt{\Sigma_{\alpha \beta }/\kappa }$.

Another curvature-elastic parameter, the spontaneous curvature, becomes crucial when the membrane segment in contact with a condensate phase forms different pattens of membrane nanotubes. The spontaneous curvature represents a quantitative measure for the transbilayer asymmetry between the two bilayer leaflets [48]. Furthermore, for engulfment of a condensate droplet by a vesicle membrane, we need to consider the line tension of the contact line, which can be positive or negative. The sign of the line tension determines the shape of the contact line and the adjacent membrane neck. A negative line tension leads to an unusual tight-lipped shape of the closed membrane neck, which prevents the fission of this neck as well as the division of the vesicle. In contrast, for a positive line tension, the neck closes in an axisymmetric manner, which can then undergo membrane fission, thereby leading to the endocytosis and uptake of the adhering condensate droplet. For nanovesicles, the sign of the contact line tension is governed by stress asymmetry between the two leaflets of the bilayer membrane [30].

This review is organized as follows. In Section 2, the different adhesion morphologies of vesicle-droplet systems will be described and their basic geometric features will be addressed in more detail. Section 3 provides several examples for aqueous solutions that generate condensate droplets. The adhesion morphologies are then characterized in terms of apparent contact angles (Section 4) and by the corresponding surface tensions, which balance each other along the contact line (Section 5). The global force balance regime and the affinity contrast *W* between the membrane and the two aqueous phases are introduced in Section 6. The subsequent Section 7 describes different morphological pathways within the force balance regime. In Section 8, the analytical theory based on curvature energies and adhesion free energies is briefly reviewed. This theory leads to a simplified expression for the affinity contrast in terms of the adhesion free energies of the two coexisting phases α and β. The curvature energy includes the bending energy, which depends on the bending rigidity κ and on the spontaneous curvature *m*. The bending rigidity leads to smoothly curved membranes and intrinsic contact angles as explained in Section 9. A large spontaneous curvature generates the formation of membrane nanotubes emanating from the vesicle membranes (Section 10). Partial and complete engulfment of droplets by vesicle membranes is discussed in Section 11, both for GUVs and for nanovesicles. The line tension λ of the contact line between the αβ interface and the membrane is examined in Section 12. The sign of this line tension determines the shape of the membrane neck that is formed during droplet engulfment (Section 13). As a consequence, a negative and positive line tension suppresses and facilitates the endocytosis and exocytosis of condensate droplets. The final Section 14 contains a summary and an outlook on open problems.

The sections are ordered according to the resolved length scales in a top-down manner, starting from the micrometer scale of giant vesicles as observed by conventional light microscopy, followed by the nanometer scale as visualized by fluorescent dyes and imaged by super-resolution STED microscopy, finally arriving at the molecular scale as studied by molecular dynamics simulations of nanovesicles. In each section, the principle of Ockham’s razor is applied by focusing on the minimal set of fluid-elastic parameters that is needed to understand a certain type of remodeling behavior. Section 4, Section 5, Section 6 and Section 7 explain the contact angles and the wetting behavior as observed for droplets adhering to giant vesicles in terms of the interfacial tension $\Sigma_{\alpha \beta }$ of the αβ interface and the affinity contrast *W* between the two aqueous phases α and β. Section 9 examines the interplay between the interfacial tension $\Sigma_{\alpha \beta }$ and the bending rigidity κ of the membrane to elucidate the mechanism for the highly curved membrane segments along the contact line and for the onset of adhesion. Section 10 focuses on the consequences of a large spontaneous curvature *m*. Finally, the line tension λ, which balances the surface tensions at the nanoscale, is introduced in Section 12 and its influence on membrane necks is described in Section 13.

## 2. Condensate Droplets Adhering to Giant Vesicles

### 2.1. Different Adhesion and Wetting Morphologies

Consider a giant vesicle which is exposed to an exterior aqueous solution that undergoes liquid–liquid phase separation into two coexisting phases, α and β as in Figure 1a. We will first consider the situation in which the condensate droplets are formed by the β phase and immersed in the bulk α phase. To this aqueous two-phase system, we add a giant unilamellar vesicle (GUV) that encloses another aqueous phase γ, which plays the role of an inert spectator phase because it does not participate in the phase separation. When such a vesicle comes into contact with one of the condensate droplets, different adhesion morphologies can be formed as shown in Figure 2.

Depending on the molecular interactions between the aqueous solutions and the vesicle membrane, the membrane may prefer the β phase over the α phase or vice versa. If the membrane strongly prefers the α phase, the whole vesicle membrane will stay in contact with this phase as depicted in Figure 2a and the β droplet will not adhere to the membrane. As a consequence, there will be no contact area between the vesicle and the droplet. The latter morphology corresponds to complete wetting by the α phase which is equivalent to complete dewetting from the β phase. On the other hand, if the membrane strongly prefers the β droplet, this droplet will spread over the whole membrane and form a thin layer on the membrane as shown in Figure 2e, which represents complete wetting by the β phase and complete dewetting from the α phase. Now, the contact area between droplet and membrane has reached its largest possible value, provided by the whole membrane area.

The intermediate morphologies in Figure 2b,d correspond to partial wetting by the α phase and partial wetting by the β droplet, respectively. Apart from the morphology in Figure 2c, all adhesion morphologies in Figure 2 can be characterized by contact angles as described further below. The morphology in Figure 2c is somewhat special because it is characterized by a flat membrane segment between the β and the γ phase, corresponding to the same pressure in both phases. Further below, we will also consider the case of balanced adhesion that is again defined in terms of the contact angles.

### 2.2. Basic Geometric Features of the Adhesion Morphologies

The three liquid phases α, β, and γ are separated by three different types of surfaces as indicated in Figure 1 and Figure 2 by different colors: the αβ interface (green dashed), the membrane segment αγ (red) exposed to the α phase, and the membrane segment βγ (purple) in contact with the β droplet. Thus, the area of the membrane segment βγ represents the contact area between vesicle and droplet. This contact area is bounded by the contact line, at which the αβ interface exerts capillary forces onto the vesicle membrane. These capillary forces lead to apparent kinks in the vesicle membranes as observed in the optical microscope, see Figure 3.

Comparison of the two morphologies in Figure 3a,b shows that the membrane segment βγ in contact with the condensate droplet β can be curved towards the γ phase within the vesicle as in Figure 3a or towards the β droplet as in Figure 3b. The sign of this curvature depends on the pressures $P_{\beta }$ and $P_{\gamma }$ within the β and γ phases. In Figure 3a, the pressure $P_{\beta }$ is smaller than the pressure $P_{\gamma }$ whereas $P_{\beta }$ exceeds $P_{\gamma }$ in Figure 3b. The pressure $P_{\gamma }$ depends on the osmotic conditions and can be changed by osmotic deflation and inflation of the vesicle.

The condensate droplets in Figure 3 arise from liquid–liquid phase separation in the aqueous solution of two synthetic polymers, PEG and dextran. This phase separation leads to the coexistence of a PEG-rich α phase and a dextran-rich β phase. The details of this phase separation will be briefly reviewed in the next section, where we discuss several systems that have been used to generate condensate droplets.

## 3. Phase Diagrams of Some Condensate-Forming Systems

### 3.1. Aqueous Two-Phase Systems

One model system for the formation of condensate droplets that has been studied in some detail is provided by aqueous solutions of the two synthetic polymers PEG and dextran. These solutions undergo segregative phase separation for relatively small weight fractions of the two polymers and represent the classic example for aqueous two-phase systems. Such systems, which are also know as aqueous biphasic systems, have been used for a long time in biochemical analysis and biotechnology and are intimately related to water-in-water emulsions [12,13].

The phase separation of the PEG-dextran solution leads to a PEG-rich phase α and to a dextran-rich phase β which coexist over a wide range of polymer concentrations. The corresponding phase diagram is displayed in Figure 4 as a function of the two weight fractions $w_{d}$ and $w_{p}$ of dextran and PEG [16]. When these two weight fractions belong to the two-phase coexistence region, the polymer solution phase separates. The compositions of the two coexisting phases correspond to the end points of the tie lines, see green dashed lines in Figure 4b. One end point describes the limit, in which the volume fraction of the PEG-rich phase α vanishes, see upward-pointing triangles in Figure 4b. The other end point of the tie line corresponds to the limit, in which the volume fraction of the dextran-rich phase β disappears, see downward-pointing triangles in Figure 4b. In general, as we move along a certain tie line, we change the volume fractions of the two coexisting phases but the interfacial tension $\Sigma_{\alpha \beta }$ of the interface between the two phases remains unchanged.

The phase diagram in Figure 4 contains a critical demixing point at $\left(w_{d},w_{p}\right)=\left(0.0451,0.0361\right)$ [9,16]. As one approaches this critical point from the two-phase coexistence region, the interfacial tension $\Sigma_{\alpha \beta }$ vanishes in a continuous manner, see Figure 5. In this figure, the distance to the critical point is measured by the deviation $\Delta c\equiv \left(c-c_{\mathrm{cr}}\right)/c_{\mathrm{cr}}$ of the total polymer concentration *c* from its critical value $c_{\mathrm{cr}}$. The interfacial tension is expected to vanish according to $\Sigma_{\alpha \beta }∼\Delta c^{\mu }$ with the critical exponent μ. This exponent has the mean field value μ=3/2 which is roughly consistent with the data in Figure 5.

Aqueous solutions of PEG and dextran represent liquid mixtures of three molecular components as given by water, PEG, and dextran. The overall phase diagram of such a three-component mixture depends on three parameters, the two weight fractions $w_{d}$ and $w_{p}$ as well as temperature. Therefore, the phase diagram in Figure 4, which was measured at room temperature, represents a two-dimensional section at constant temperature across the full three-dimensional phase diagram.

The aqueous phase separation of PEG-dextran solutions provides an example for segregative separation, in which one phase is enriched in one macromolecular component whereas the other phase is enriched in the other macromolecular component. This segregative behavior implies that the different species of macromolecules effectively repel each other. Another type of aqueous two-phase system is obtained by associative phase separation, in which one phase is enriched in the macromolecular components whereas the other phase represents a dilute macromolecular solution [19,20,21,22]. The associative behavior implies that the different macromolecular species effectively attract each other. Such behavior is observed, for instance, in solutions of two polyelectrolytes that are oppositely charged. The latter type of phase separation is also known as coacervation and leads to coacervate droplets enriched in the polyelectrolytes. Phase diagrams for associative phase separation of polyelectrolyte solutions have been measured for a variety of polyelectrolytes and are typically displayed as a function of polymer and salt concentrations for constant temperature. The latter phase diagrams typically exhibit a one-phase region at high salt concentration and a two-phase coexistence region at low salt concentration [21,22]. In the context of fluid elasticity, coacervate droplets represent a special kind of condensate droplets.

### 3.2. Binary Liquid Mixture in Silico

A relatively simple model system that leads to the formation of condensate droplets is provided by a binary mixture consisting of water and solute molecules. The mixture is modeled in terms of water (W) and solute (S) beads, both of which represent small molecular groups. For computational simplicity, the two types of beads are taken to have the same size and the interaction between two W beads is taken to be the same as the interaction between two S beads [29,30,50]. This symmetry implies that the phase diagram does not change when we substitute the W by the S beads and that this binary mixture has a particularly simple phase diagram as displayed in Figure 6. This binary mixture represents an off-lattice variant of the classical lattice gas model for binary mixtures.

The phase diagram in Figure 6 involves two coordinates, the solute mole fraction $\Phi_{S}$ and the solubility ζ of the solute molecules in water. The mole fractions $\Phi_{S}$ and $\Phi_{W}$ of solute and water are defined by
(1)$$\Phi_{S}=\frac{N_{S}}{N_{W}+N_{S}}\;\mathrm{and}\;\Phi_{W}=\frac{N_{W}}{N_{W}+N_{S}}=1-\Phi_{S}$$
where $N_{S}$ and $N_{W}$ are the numbers of S and W beads. The solubility is defined in terms of the interaction parameters between the W and S beads [50]. The solubility plays the same role as the temperature of the vesicle-droplet system. Inspection of Figure 6 shows that the phase diagram is mirror symmetric with respect to $\Phi_{S}=1/2$. This symmetry implies horizontal tie lines, which are parallel to the $\Phi_{S}$-axis. The symmetry also implies that the critical demixing point is located at $\Phi_{S}=\Phi_{W}=1/2$. The phase diagram in Figure 6 is qualitatively similar to the phase diagrams obtained from mean field theories but represents the result of extensive molecular dynamics simulations.

### 3.3. Phase Behavior of Protein Condensates

Recently, condensate droplets enriched in certain proteins have also been observed within living cells. These condensates represent membraneless organelles and behave like liquid droplets. Examples for these kinds of condensates include germ P-bodies [31], nucleoli [33], and stress granules [34], as reviewed in ref [35]. These protein condensates are believed to form via liquid–liquid phase separation in the cytoplasm and can be reconstituted in vitro [36,37,38,39]. They are enriched in certain types of proteins that have intrinsically disordered domains and interact via multivalent macromolecular interactions [35,38,39,40,41]. The phase behavior of protein condensates has been studied as a function of protein concentration and temperature, both in vivo [32] and in vitro [51]. One example for an in-vivo phase diagram is displayed in Figure 7.

## 4. Contact Angles of Adhesion Morphologies

### 4.1. Apparent Versus Intrinsic Contact Angles

In order to analyze the adhesion morphologies in Figure 1, Figure 2 and Figure 3 in a quantitative manner, we need to consider the contact angles along the contact line between the droplet and the vesicle membrane. We will describe this analysis in two steps. First, we will discuss those contact angles that can be resolved by conventional light microscopy as in Figure 3. These contact angles are *apparent* contact angles because they are defined with respect to the apparent membrane kink at the contact line. However, if such a kink persisted to nanoscopic length scales, the membrane would acquire a very large bending energy. Therefore, each kink in Figure 3 should be replaced by a smoothly curved membrane segment on sufficiently small length scales [5]. Such smoothly curved segments have indeed been observed by super-resolution STED microscopy [52] as described further below.

Second, we will describe the intrinsic contact angles which are a direct consequence of the requirement that the membrane should be smoothly curved on the nanometer scale. In order to simplify the discussion in the present and the following sections, we will often use the shorter term ‘contact angle’ as an abbreviation of ‘apparent contact angle’.

### 4.2. Different Wetting Regimes from Apparent Contact Angles

The vesicle-droplet morphologies in Figure 1, Figure 2 and Figure 3 can be analyzed in terms of three apparent contact angles $\theta_{\alpha },\theta_{\beta }$, and $\theta_{\gamma }$ as shown in Figure 8. The contact angle $\theta_{\alpha }$ is the angle between the αβ interface and the αγ membrane segment, the contact angle $\theta_{\beta }$ represents the angle between the αβ interface and the βγ segment, and the contact angle $\theta_{\gamma }$ is the angle between the βγ and the αγ membrane segments. More precisely, these angles are defined with respect to the tangent planes of these three surfaces at the contact line.

Inspection of Figure 8 shows that the three apparent contact angles satisfy the obvious relation
(2)$$\theta_{\alpha }+\theta_{\beta }+\theta_{\gamma }=2\pi =360^{\circ }\;.$$Therefore, the value of the contact angle $\theta_{\gamma }$ is determined by the values of the two contact angles $\theta_{\alpha }$ and $\theta_{\beta }$. As a consequence, we can characterize the different adhesion morphologies in Figure 2 and Figure 3 by the relative size of $\theta_{\alpha }$ and $\theta_{\beta }$. Furthermore, it is important to note that the contact angles are *local* properties of the vesicle-droplet morphology which characterize the local vicinity of the contact line. This local viewpoint is emphasized in Figure 9 which displays the local vicinity of the contact lines for all morphologies in Figure 2.

The limiting case with $\theta_{\alpha }=0$ as well as $\theta_{\beta }=\pi$ and $\theta_{\gamma }=\pi$ in Figure 9a describes complete wetting of the membrane by the α phase or, equivalently, complete dewetting of the membrane from the β droplet. Likewise, the limiting case with $\theta_{\beta }=0$ as well as $\theta_{\alpha }=\pi$ and $\theta_{\gamma }=\pi$ in Figure 9e corresponds to complete wetting of the membrane by the β droplet or, equivalently, to complete dewetting of the membrane from the α phase. All five wetting regimes illustrated in Figure 9 are also included in Table 1. In both Figure 9 and Table 1, we introduced the additional regime of balanced adhesion with $\theta_{\alpha }=\theta_{\beta }$.

As previously mentioned, the contact angles are *local* properties of the vesicle-droplet morphology, which implies that they do not determine the overall adhesion morphology. Indeed, the contact angles remain unchanged when we rotate the αβ interface and the two membrane segments around the contact line, which implies that the overall orientation of the contact angles involves one rotation angle which is determined by *global* properties such as the vesicle volume and the droplet volume.

## 5. Balance of Surface Tensions along Contact Line

### 5.1. From Apparent Contact Angles to Surface Tensions

The contact angles $\theta_{\alpha }$, $\theta_{\beta }$, and $\theta_{\gamma }$ are the angles between the tangent planes of the αβ interface and of the two membrane segments at the contact line, see Figure 9. Each of these three surfaces is subject to a certain mechanical tension as provided by the interfacial tension $\Sigma_{\alpha \beta }$ as well as by the mechanical tensions $\Sigma_{\alpha \gamma }^{m}$ and $\Sigma_{\beta \gamma }^{m}$ of the two membrane segments αγ and βγ. Note that the segment tensions have a superscript ‘m’ which stands for ‘membrane’. This distinction is necessary because the interfacial tension $\Sigma_{\alpha \beta }$ is a material parameter whereas the mechanical tensions of the membrane segments depend both on the size and on the shape of these segments.

### 5.2. Triangle Formed by Three Surface Tensions

Mechanical equilibrium of the vesicle-droplet system implies that the interfacial tension $\Sigma_{\alpha \beta }$ has a constant value for the whole αβ interface. Likewise, the tensions $\Sigma_{\alpha \gamma }^{m}$ and $\Sigma_{\beta \gamma }^{m}$ are also constant everywhere on the αγ and βγ membrane segments. Furthermore, mechanical equilibrium also implies that the contact line does not move and that the two segment tensions $\Sigma_{\alpha \gamma }^{m}$ and $\Sigma_{\beta \gamma }^{m}$ are balanced by the interfacial tension $\Sigma_{\alpha \beta }$. This force balance is illustrated in Figure 10a for the case of partial wetting by the β droplet.

The force balance in Figure 10a implies that the three surface tensions form a triangle as shown in Figure 10b [8,28,53]. Such a force balance is also possible in a liquid mixture with three coexisting liquid phases [28]. In the latter case, a β and a γ droplet may adhere to each other and coexist with the liquid bulk phase α which then leads to a force balance between the three interfacial tensions $\Sigma_{\alpha \beta }$, $\Sigma_{\beta \gamma }$, and $\Sigma_{\alpha \gamma }$ along the three-phase contact line, The corresponding triangle of the three interfacial tensions is known as Neumann’s triangle [54].

It is instructive to consider the tension triangle for the other wetting regimes as well. For complete wetting by the α phase as shown in Figure 9a, we then obtain a degenerate triangle with $\theta_{\alpha }=0$ or
(3)$$\Sigma_{\beta \gamma }^{m}=\Sigma_{\alpha \gamma }^{m}+\Sigma_{\alpha \beta }\;(\mathrm{complete}\;\mathrm{wetting}\;\mathrm{by}\;\alpha ).$$For partial wetting by the α phase as in Figure 9b, we have $0<\theta_{\alpha }<\theta_{\beta }$ which implies
(4)$$\Sigma_{\beta \gamma }^{m}<\Sigma_{\alpha \gamma }^{m}+\Sigma_{\alpha \beta }\;(\mathrm{partial}\;\mathrm{wetting}\;\mathrm{by}\;\alpha ).$$For balanced adhesion as in Figure 9c, the tension triangle becomes an isosceles triangle with equal contact angles $\theta_{\alpha }=\theta_{\beta }$ and equal tensions
(5)$$\Sigma_{\beta \gamma }^{m}=\Sigma_{\alpha \gamma }^{m}\;(\mathrm{balanced}\;\mathrm{adhesion})$$
of the two membrane segments. Inspection of the tension triangle in Figure 10b, corresponding to Figure 9c, shows that
(6)$$\Sigma_{\alpha \gamma }^{m}<\Sigma_{\beta \gamma }^{m}+\Sigma_{\alpha \beta }\;(\mathrm{partial}\;\mathrm{wetting}\;\mathrm{by}\;\beta ).$$Finally, for complete wetting by the β phase as in Figure 9e, we obtain another degenerate triangle with $\theta_{\beta }=0$ or
(7)$$\Sigma_{\alpha \gamma }^{m}=\Sigma_{\beta \gamma }^{m}+\Sigma_{\alpha \beta }\;(\mathrm{complete}\;\mathrm{wetting}\;\mathrm{by}\;\beta ).$$All tension-tension relationships as given by Equations (Equation 3)–(Equation 7) follow from the simple and general property of triangles that each side of a triangle must be smaller than or equal to the sum of the two other sides. These relationships are summarized in Table 2.

### 5.3. General Consequences of the Tension Triangle

The interfacial tension $\Sigma_{\alpha \beta }$ is always positive as required by thermodynamic stability. In principle, the tensions $\Sigma_{\alpha \gamma }^{m}$ and $\Sigma_{\beta \gamma }^{m}$ of the membrane segments αγ and βγ can be positive or negative, corresponding to a stretched or compressed membrane segment. However, when the three surface tensions balance each other as in Figure 10, the two segment tensions must be positive as well, i.e.,
(8)$$\Sigma_{\alpha \gamma }^{m}>0\;\mathrm{and}\;\Sigma_{\beta \gamma }^{m}>0\;.$$Furthermore, each internal angle $\eta_{i}=\pi -\theta_{i}$ of the tension triangle must satisfy $0\leq \eta_{i}\leq \pi =180^{\circ }$ which implies the inequalities
(9)$$0\leq \theta_{i}\leq \pi =180^{\circ }\;\mathrm{with}\;i=\alpha ,\beta \;\mathrm{and}\;\gamma ,$$
for the external angles $\theta_{i}$, which are equal to the apparent contact angles. Therefore, the force balance of the three surface tensions at the contact line implies that $\pi =180^{\circ }$ is the largest possible value of these contact angles.

### 5.4. Limit of Small Contact Angle $\theta_{\gamma }$

The tension-tension relationships described by Equations (Equation 3)–(Equation 7) and Table 2 are obtained from the triangle inequalities $\Sigma_{\alpha \gamma }^{m}\leq \Sigma_{\beta \gamma }^{m}+\Sigma_{\alpha \beta }$ and $\Sigma_{\beta \gamma }^{m}\leq \Sigma_{\alpha \gamma }^{m}+\Sigma_{\alpha \beta }$, i.e., by focusing on the two sides $\Sigma_{\alpha \gamma }^{m}$ and $\Sigma_{\beta \gamma }^{m}$ of the tension triangle in Figure 10b. The two equalities $\Sigma_{\alpha \gamma }^{m}=\Sigma_{\beta \gamma }^{m}+\Sigma_{\alpha \beta }$ and $\Sigma_{\beta \gamma }^{m}=\Sigma_{\alpha \gamma }^{m}+\Sigma_{\alpha \beta }$ are then obtained in the limits of small $\theta_{\alpha }$ and small $\theta_{\beta }$, corresponding to complete wetting by the α and by the β phase, respectively. It is also possible for the third contact angle $\theta_{\gamma }$ to become small. The corresponding tension-tension relationship has the form
(10)$$\Sigma_{\alpha \beta }\leq \Sigma_{\alpha \gamma }^{m}+\Sigma_{\beta \gamma }^{m}\;,$$
which represents the triangle inequality for the side $\Sigma_{\alpha \beta }$ of the tension triangle in Figure 10b. The limit of small apparent contact angle $\theta_{\gamma }$ now leads to
(11)$$\Sigma_{\alpha \beta }=\Sigma_{\alpha \gamma }^{m}+\Sigma_{\beta \gamma }^{m}\;\mathrm{for}\;\theta_{\gamma }=0.$$This limit applies to complete engulfment of the β droplet by the vesicle membrane as displayed in Figure 3c. Because the relation in Equation (Equation 11) is symmetric when we permute the two liquid phases α and β, the same relation applies to the complete engulfment of an α droplet by the vesicle membrane. These engulfment processes are important because they represent the first step of droplet endocytosis and exocytosis by the vesicle as will be discussed in more detail further below.

## 6. Balanced Surface Tensions and Affinity Contrast

### 6.1. Force Balance Regime for Surface Tensions

A combination of Equations (Equation 3) and (Equation 4) leads to the tension-tension relationship
(12)$$\Sigma_{\beta \gamma }^{m}\leq \Sigma_{\alpha \gamma }^{m}+\Sigma_{\alpha \beta }\;\mathrm{or}\;\Sigma_{\beta \gamma }^{m}-\Sigma_{\alpha \gamma }^{m}\leq \Sigma_{\alpha \beta }$$
for complete and partial wetting by the α phase. Likewise, a combination of Equations (Equation 6) and (Equation 7) leads to
(13)$$\Sigma_{\alpha \gamma }^{m}\leq \Sigma_{\beta \gamma }^{m}+\Sigma_{\alpha \beta }\;\mathrm{or}\;-\Sigma_{\alpha \beta }\leq \Sigma_{\beta \gamma }^{m}-\Sigma_{\alpha \gamma }^{m}$$
for complete and partial wetting by the β phase. Finally, the two relations for the tension difference $\Sigma_{\beta \gamma }^{m}-\Sigma_{\alpha \gamma }^{m}$ as given by Equations (Equation 12) and (Equation 13) imply
(14)$$-\Sigma_{\alpha \beta }\leq \Sigma_{\beta \gamma }^{m}-\Sigma_{\alpha \gamma }^{m}\leq +\Sigma_{\alpha \beta }\;\mathrm{or}\;-1\leq \frac{\Sigma_{\beta \gamma }^{m}-\Sigma_{\alpha \gamma }^{m}}{\Sigma_{\alpha \beta }}\leq +1\;.$$Multiplying these inequalities by −1, we conclude that the same inequalities hold for the tension difference $\Sigma_{\alpha \gamma }^{m}-\Sigma_{\beta \gamma }^{m}$ as well.

### 6.2. Affinity Contrast between Coexisting Liquid Phases

We now define the affinity contrast between the two coexisting liquid phases α and β via [28]
(15)$$W\equiv \Sigma_{\beta \gamma }^{m}-\Sigma_{\alpha \gamma }^{m}\;.$$The affinity contrast *W* is negative if the membrane prefers the β phase over the α phase and positive if the membrane prefers the α phase over the β phase. It then follows from Equation (Equation 14) that the affinity contrast *W* satisfies the inequalities
(16)$$-\Sigma_{\alpha \beta }\leq W\leq +\Sigma_{\alpha \beta }$$
which implies the inequalities
(17)−1≤w≤+1
for the rescaled affinity contrast
(18)$$w\equiv \frac{W}{\Sigma_{\alpha \beta }}=\frac{\Sigma_{\beta \gamma }^{m}-\Sigma_{\alpha \gamma }^{m}}{\Sigma_{\alpha \beta }}\;.$$The limiting case w=−1 describes complete wetting of the vesicle membrane by the β phase and complete dewetting of the membrane from the α phase, compare Table 2. Likewise, the limiting case w=+1 describes complete wetting of the vesicle membrane by the α phase and complete dewetting of the membrane from the β phase.

We can visualize the inequalities in Equation (Equation 17) by the yellow force balance regime in Figure 11, where the three surface tensions can balance each other. The two coordinates *x* and *y* used in Figure 11 are defined by
(19)$$x\equiv \frac{\Sigma_{\alpha \gamma }^{m}}{\Sigma_{\alpha \beta }}\;\mathrm{and}\;y\equiv \frac{\Sigma_{\beta \gamma }^{m}}{\Sigma_{\alpha \beta }}\;,$$
corresponding to the membrane segment tensions $\Sigma_{\alpha \gamma }^{m}$ and $\Sigma_{\beta \gamma }^{m}$ divided by the interfacial tension $\Sigma_{\alpha \beta }$. The force balance regime is bounded from below by the line of complete wetting by the β phase and from above by the line of complete wetting by the α phase.

The third boundary in Figure 11, which truncates the force balance regime for small values of *x* and *y*, corresponds to the limit of small values for the apparent contact angle $\theta_{\gamma }$ which leads to $\Sigma_{\alpha \beta }=\Sigma_{\alpha \gamma }^{m}+\Sigma_{\beta \gamma }^{m}$ as in Equation (Equation 11) or to
(20)$$y=\frac{\Sigma_{\beta \gamma }^{m}}{\Sigma_{\alpha \beta }}=1-\frac{\Sigma_{\alpha \gamma }^{m}}{\Sigma_{\alpha \beta }}=1-x\;\mathrm{in}\;\mathrm{terms}\;\mathrm{of}\;x\;\mathrm{and}\;y.$$For three coexisting liquid phases, this latter relationship would describe complete wetting by the γ phase. In the present context, the relationship in Equation (Equation 20) corresponds to complete engulfment of a condensate droplet by the vesicle membrane. This droplet may be formed by the β phase as in Figure 3c or by the α phase, depending on the sign of the affinity contrast *W*.

Each triple of surface tensions $\Sigma_{\alpha \beta }$, $\Sigma_{\alpha \gamma }^{m}$, and $\Sigma_{\beta \gamma }^{m}$ leads to a unique point (x,y) in Figure 11. All tension triples which are located outside of the yellow force balance regime cannot balance each other and, thus, cannot belong to an adhesion morphology with a stable contact line between vesicle membrane and droplet. More precisely, all points (x,y) that are located in Figure 11 below the line of complete wetting (CWβ) by the β phase represent vesicles that avoid any contact with the α phase as in Figure 2e, and all points (x,y) in Figure 11 above the line of complete wetting (CWα) by the α phase describe vesicles without any contact to the β droplet as in Figure 2a.

The force balance regime in Figure 11 contains the corner point with x=1 and y=0, which corresponds to vanishing tension $\Sigma_{\beta \gamma }^{m}$ of the βγ membrane segment, for which the interfacial tension $\Sigma_{\alpha \beta }$ is only balanced by the tension $\Sigma_{\alpha \gamma }^{m}$ of the αγ segment as follows from the tension triangle in Figure 10b. In this limiting case, the contact angle $\theta_{\alpha }$ approaches $\pi =180^{\circ }$ irrespective of the contact angles $\theta_{\beta }$ and $\theta_{\gamma }$. Likewise, the second corner point with x=0 and y=1 in Figure 11 corresponds to vanishing tension $\Sigma_{\alpha \gamma }^{m}$ within the αγ membrane segment, in which the interfacial tension $\Sigma_{\alpha \beta }$ is only balanced by the tension $\Sigma_{\beta \gamma }^{m}$ of the βγ segment. In the latter limit, the contact angle $\theta_{\beta }$ becomes close to $\pi =180^{\circ }$ irrespective of the contact angles $\theta_{\alpha }$ and $\theta_{\gamma }$ as follows again from the tension triangle.

### 6.3. Relation between Affinity Contrast and Apparent Contact Angles

When the vesicle-droplet morphology exhibits a non-moving contact line, the mechanical equilibrium is characterized by a tension triangle as in Figure 10. Such a triangle also implies simple and general relations between the surface tensions and the apparent contact angles as follows from the law of sines for triangles. This law states that the ratio of any two sides of a triangle is equal to the ratio of the sines for the two internal angles that are opposite to these two sides. As before, we denote the internal angles of the tension triangle in Figure 10b by
(21)$$\eta_{i}=\pi -\theta_{i}\;\mathrm{for}\;i=\alpha ,\beta ,\;\mathrm{and}\;\gamma .$$The law of sines then leads to the equalities
(22)$$\frac{\Sigma_{\alpha \gamma }^{m}}{\Sigma_{\alpha \beta }}=\frac{\sin\eta_{\beta }}{\sin\eta_{\gamma }}=\frac{\sin\theta_{\beta }}{\sin\theta_{\gamma }}\;\mathrm{and}\;\frac{\Sigma_{\beta \gamma }^{m}}{\Sigma_{\alpha \beta }}=\frac{\sin\eta_{\alpha }}{\sin\eta_{\gamma }}=\frac{\sin\theta_{\alpha }}{\sin\theta_{\gamma }}\;.$$By taking the ratio of these two equations, we also obtain the relation
(23)$$\frac{\Sigma_{\alpha \gamma }^{m}}{\Sigma_{\beta \gamma }^{m}}=\frac{\sin\theta_{\beta }}{\sin\theta_{\alpha }}\;.$$Therefore, the tensions $\Sigma_{\alpha \gamma }^{m}$ and $\Sigma_{\beta \gamma }^{m}$ of the two membrane segments are equal to each other for equal contact angles $\theta_{\beta }$ and $\theta_{\alpha }$, which corresponds to balanced adhesion as in Figure 9c and in the third row of Table 2.

We now take the difference of the two equalities in Equation (Equation 22) to obtain [8,28]
(24)$$\frac{\Sigma_{\beta \gamma }^{m}-\Sigma_{\alpha \gamma }^{m}}{\Sigma_{\alpha \beta }}=\frac{\sin\theta_{\alpha }-\sin\theta_{\beta }}{\sin\theta_{\gamma }}=w$$
where the second equality follows from the definition of the rescaled affinity contrast *w* in terms of the three surface tensions as given by Equation (Equation 18). Therefore, the rescaled affinity contrast *w*, which is a mechanical quantity, is directly related to the three apparent contact angles, which are purely geometric quantities and can be measured by light microscopy, see Figure 3. The affinity contrast $W=w\Sigma_{\alpha \beta }$ is then obtained by multiplying *w* with the interfacial tension $\Sigma_{\alpha \beta }$ which represents a material parameter that can be measured as well, see Figure 5.

## 7. Transitions between Different Wetting Morphologies

So far, we discussed the different adhesion geometries in Figure 2 and distinguished these morphologies by the apparent contact angles as well as by the three surface tensions and the resulting affinity contrast *W*. In what follows, we will now consider possible transitions between these morphologies that can be obtained by changing a certain control parameter of the vesicle-droplet system. Both the contact angles and the surface tensions reflect the underlying molecular interactions which can be varied by changes in the molecular composition of vesicle membrane and liquid phases as well as by temperature and osmotic pressure. Such a variation leads to a certain morphological pathway that can be visualized in the parameter space of Figure 11.

### 7.1. Different Morphological Pathways

The morphology diagram in Figure 11 is defined in terms of the two r tension ratios $x=\Sigma_{\alpha \gamma }^{m}/\Sigma_{\alpha \beta }$ and $y=\Sigma_{\beta \gamma }^{m}/\Sigma_{\alpha \beta }$, corresponding to the tensions $\Sigma_{\alpha \gamma }^{m}$ and $\Sigma_{\beta \gamma }^{m}$ of the two membrane segments divided by the interfacial tension $\Sigma_{\alpha \beta }$. As we change a control parameter that affects these three tensions, we move in this parameter space along a one-dimensional pathway as illustrated in Figure 12. The green morphological pathway in this figure leads to complete engulfment of the β droplet by the membrane as displayed in Figure 3c; the red pathway will be discussed in some detail further below; and the purple pathway starts from complete wetting by the α phase and ends up with complete wetting by the β phase, thereby crossing the whole force balance region. If we followed the latter pathway, we would sequentially observe all adhesion morphologies displayed in Figure 2, from the left-most morphology with no adhesion of vesicle and β droplet to the right-most morphology, for which the vesicle membrane is completely covered by the β phase.

The morphological pathways displayed in Figure 12 can be induced by several experimental procedures. A relatively simple procedure is osmotic deflation of the vesicle by increasing the osmolarity in the exterior compartment, which acts to decrease the tensions $\Sigma_{\alpha \gamma }^{m}$ and $\Sigma_{\beta \gamma }^{m}$ of the two membrane segments. The latter procedure led to the complete engulfment morphology in Figure 3c. Another simple procedure is available for two-phase systems within giant vesicles. When such a vesicle is exposed to osmotic deflation, the polymer concentration is increased within the vesicle, thereby moving the aqueous two-phase system deeper into the two-phase coexistence region as described in the next subsection.

### 7.2. Complete-to-Partial Wetting Transitions

A complete-to-partial wetting transition was first observed for phase-separated PEG-dextran solutions within GUVs [6]. In this first study, the GUV membranes were composed of 95 mol % of the phospholipid DOPC and doped with 4 mol % of the glycolipid GM1. Analogous complete-to-partial wetting transitions were also observed for ternary lipid mixtures consisting of two phospholipids, DOPC and DPPC, as well as cholesterol [9]. In this second study, two different compositions of this ternary mixture were studied, corresponding to lipid bilayers in the liquid-disordered and the liquid-ordered lipid phase [55,56,57].

For all three lipid compositions, the wetting behavior of the PEG-dextran solutions was observed to be quite similar as schematically shown in Figure 13. The two-phase region above the binodal line in Figure 13a is divided up into two subregions, corresponding to a complete wetting (CW) subregion close to the critical point and a partial wetting (PW) subregion further away from this point. The boundary between the CW and PW subregions is provided by a certain tie line, the location of which depends on the composition of the lipid membranes. Within the CW subregion, the vesicle membrane is completely wetted by the PEG-rich phase α and has no contact with the dextran-rich β phase, see Figure 13b. Within the PW subregime, the membrane is in contact with both liquid phases α and β and forms a contact line with the αβ interface, see Figure 13c.

As described further below, the membrane segment αγ in contact with the PEG-rich phase α acquires a large spontaneous curvature which leads to the formation of many membrane nanotubes that protrude into the PEG-rich phase within the GUVs. Refs. [7,8,9] These nanotubes have a width below the spatial resolution of conventional fluorescence microscopy but are still visible because of the fluorescently labeled membranes. For polymer concentrations that belong to the CW subregion of the two-phase coexistence region, the nanotubes are completely immersed in the PEG-rich phase α and avoid any contact with the dextran-rich phase β. For larger polymer concentrations corresponding to the PW subregion, the nanotubes adhere to the αβ interface between the two liquid phases α and β. Therefore, the behavior of the membrane nanotubes can be used to distinguish between the CW and the PW subregions.

The dashed tie line in Figure 13a, which provides the boundary between the CW and the PW subregions, also partitions the binodal line into two line segments, which are colored red and blue in this figure. If one approaches the red segment of the binodal line from the one-phase region, a wetting layer of the α phase starts to build up at the membrane and becomes mesoscopically thick as one reaches this line segment. No such layer is formed along the blue segment of the binodal line. More precisely, the phase diagram shown in Figure 13a applies to a continuous or second-order transition from complete to partial wetting. If this transition is discontinuous or first-order, the boundary point between the red and blue segments of the binodal becomes a prewetting line that extends into the one-phase region below the binodal line. Along the prewetting line, one observes a transition from a relatively thick to a relatively thin wetting layer. In the context of wetting by condensates, prewetting behavior has been recently studied [58] using a Landau-type model for semi-infinite systems [59,60].

### 7.3. Vesicle-Droplet Systems with Two Wetting Transitions

In general, it should be possible to modify the molecular interactions between PEG, dextran, and the lipid bilayers in order to obtain a partial-to-complete wetting transition by the dextran-rich β phase as well. Combining such a transition with the partial-to-complete wetting transition by the PEG-rich phase as described by Figure 13 would provide a morphological pathway that resembles the purple pathway in Figure 12. If we were able to move the vesicle-droplet system along such a purple pathway by changing a single control parameter, we would observe two subsequent wetting transitions in the same system.

So far, no such control parameter has been found for vesicle-droplet systems that involve aqueous two-phase systems of PEG and dextran. On the other hand, for condensate droplets that are enriched in the soybean protein glycinin [51], several such control parameters have been recently identified [61]. One such control parameter is the salt concentration in the aqueous buffer. Increasing the salt concentration from low to intermediate values, the vesicle-droplet system undergoes a complete-to-partial dewetting transition whereas a further increase from intermediate to large salt concentrations leads to a partial-to-complete wetting transition of the glycinin-rich droplets at the vesicle membranes. This behavior strongly indicates that electrostatic interactions play an important role for membranes exposed to glycinin-rich condensates.

Wetting transitions of two coexisting phases in contact with a solid substrate or a macroscopic liquid–liquid interface have been studied for a long time [59,62,63,64,65] but no such system has been previously described, to the best of my knowledge, that undergoes two distinct wetting transitions at constant temperature. On the other hand, electrostatic interactions are also crucial for aqueous two-phase systems that are formed in solutions of oppositely charged polyelectrolytes by associative phase separation. Electrostatic interactions will always be affected by changes in the salt concentration, which provides another control parameter for the phase behavior. Thus, it is likely that vesicles interacting with coacervate droplets containing oppositely charged polyelectrolytes will exhibit two wetting transitions as well.

## 8. Theory of Curvature Elasticity and Vesicle-Droplet Adhesion

### 8.1. Fine Structure of Apparent Membrane Kinks

Using a conventional optical microscope, one typically observes adhesion morphologies with apparent kinks of the vesicle membranes as in Figure 3. However, if such a kink persisted to nanoscopic length scales, the membrane would acquire a very large bending energy. It is thus plausible to assume that the kinks in Figure 3 will be replaced by a smoothly curved membrane segment on sufficient small length scales [5]. This assumption has been recently confirmed by super-resolution STED microscopy [52], see Figure 14. In what follows, we will assume that all kinks in Figure 3 will be smoothened out when observed with sufficiently high resolution.

### 8.2. Curvature and Curvature Elasticity of Membranes

On length scales which are somewhat larger than the membrane thickness, we can describe the membrane surface as a smoothly curved surface as follows from the shape fluctuations observed in molecular dynamics simulations [66]. We can then apply the mathematical concepts of differential geometry to such a membrane surface. Each point of a smoothly curved surface defines two principal curvatures, $C_{1}$ and $C_{2}$ [67], which are local quantities that vary along the membrane surface. Using the two principal curvatures $C_{1}$ and $C_{2}$, the mean curvature is defined by
(25)$$M=\frac{1}{2}\left(C_{1}+C_{2}\right)$$
and the Gaussian curvature by
(26)$$G=C_{1}\;C_{2}\;.$$The principal curvatures $C_{1}$ and $C_{2}$ as well as the mean curvature *M* and the Gaussian curvature *G* are geometric quantities that do not depend on the choice of the surface coordinates, i.e., they are invariants under the reparametrization of the membrane surface [68]. In the mathematical literature, the mean curvature *M* is often denoted by the symbol *H* and the Gaussian curvature *G* by the symbol *K*.

The curvature elasticity of a membrane introduces three curvature-elastic parameters: the bending rigidity κ, which governs the resistance of the membrane against bending deformations; the spontaneous curvature *m*, which represents the preferred curvature of the membrane; and the Gaussian curvature modulus $\kappa_{G}$, which becomes important when the membrane undergoes topological transformations. The spontaneous curvature *m* takes into account that all biomembranes are built up from two leaflets of lipid molecules and that these two leaflets may have different densities and compositions. Another contribution to the spontaneous curvature arises from the asymmetry between the interior and exterior aqueous solution which leads to different molecular interactions of these solutions with the outer and inner leaflets of the bilayer membranes.

In the framework of the spontaneous curvature model, the elastic curvature energy of the membrane is given by the area integral [68,69,70]
(27)$$E_{\mathrm{cu}}=\int \;dA\;\left[2\kappa {\left(M-m\right)}^{2}+\kappa_{G}G\right]\;.$$For a closed vesicle without membrane edges or pores, the Gauss-Bonnet theorem of differential geometry implies that the Gaussian curvature energy $E_{G}$ has the form
(28)$$E_{G}\equiv \int \;dA\;\kappa_{G}G=2\pi \chi \kappa_{G}=2\pi \left(2-2g\right)\kappa_{G}$$
where χ is the Euler characteristic and g the topological genus, which counts the number of handles formed by the closed surface [67]. Both the Euler characteristic and the topological genus have a constant value as long as the vesicle does not change its topology. Therefore, in the absence of topological transformations, the constant energy term proportional to the Gaussian curvature modulus $\kappa_{G}$ can be ignored and the curvature energy reduces to the elastic bending energy
(29)$$E_{\;\mathrm{be}}=\int \;dA\;2\kappa {\left(M-m\right)}^{2}$$
which becomes small when the mean curvature *M* is close to the spontaneous curvature *m*. At the end of this paper, we will consider the process of droplet endocytosis which involves the division of a vesicle into two daughter vesicles, thereby changing the membrane topology.

### 8.3. Shapes of Giant Vesicles in the Absence of Condensate Droplets

In the absence of condensate droplets, the experimentally observed vesicle shapes can be obtained by minimizing the bending energy $E_{\;\mathrm{be}}$ in Equation (Equation 29), provided one takes additional constraints on the membrane area and the vesicle volume into account. At constant temperature, the membrane area of lipid bilayers is constant, reflecting the ultralow solubility of the lipid molecules. Likewise, the volume of the vesicle is conserved for constant pressure difference
(30)$$\Delta P=P_{\mathrm{ex}}-P_{\mathrm{in}}$$
between the pressures $P_{\mathrm{in}}$ and $P_{\mathrm{ex}}$ of the interior and exterior solutions, which requires constant osmotic conditions. We are then led to minimize the vesicle’s shape functional [70,71]
(31)$$F_{\mathrm{ve}}=\Delta PV+\Sigma A+E_{\;\mathrm{be}}=-\Delta PV+\Sigma A+2\;\kappa \int \;\;dA\;{\left(M-m\right)}^{2}$$
and to treat the parameters ΔP and Σ as Lagrange multipliers that allow us to perform the constrained minimization of the bending energy for a certain vesicle volume *V* and a certain membrane area *A*. Several recent studies have demonstrated that the shapes of GUVs calculated in this manner agree quantitatively with the experimentally observed shapes [72,73]. In these latter experiments, the lipid membranes contained cholesterol which undergoes frequent flip-flops and implies that area-difference elasticity [74,75,76] plays no role, which is useful because the latter type of elasticity would otherwise introduce two additional parameters.

The physical meaning of the Lagrange multiplier tension Σ has been unclear for many years but turns out to have a very simple physical interpretation, directly related to the stretching (and compression) energy
(32)$$E_{\mathrm{st}}=\frac{K_{A}}{2}\frac{{\left(A-A_{0}\right)}^{2}}{A_{0}}$$
and the associated mechanical tension
(33)$$\Sigma_{\mathrm{st}}=\frac{dE_{\mathrm{st}}}{dA}=K_{A}\frac{A-A_{0}}{A_{0}}$$
which are both proportional to the area compressibility modulus $K_{A}$. The membrane is tensionless when the membrane area *A* attains its optimal value $A_{0}$. Using a two-step procedure for the minimization of the combined bending and stretching energy $E_{\;\mathrm{be}}+E_{\mathrm{st}}$ for fixed volume *V*, one can show that $\Sigma =\Sigma_{\mathrm{st}}$, i.e., the Lagrange multiplier tension Σ, which ensures that the area has the prescribed value *A*, is equal to the mechanical tension $\Sigma_{\mathrm{st}}$, which was generated by increasing the membrane area from $A_{0}$ to *A* [77]. Thus, we do not need to distinguish the two tensions by different symbols and will denote both of them by Σ.

### 8.4. Shape Functional of Vesicle-Droplet System

For the vesicle-droplet systems, we have to include the interfacial free energy $\Sigma_{\alpha \beta }A_{\alpha \beta }$ of the αβ interface with area $A_{\alpha \beta }$ as well as the adhesion free energies of the αγ and βγ membrane segments, in addition to the the bending and stretching energies of the vesicle membrane. The adhesion free energies are proportional to the surface areas $A_{\alpha \gamma }$ and $A_{\beta \gamma }$ of the two membrane segments with the total surface area *A* of the vesicle membrane given by
(34)$$A=A_{\alpha \gamma }+A_{\beta \gamma }\;.$$The corresponding adhesion free energies per unit area are taken to be $W_{\alpha \gamma }$ and $W_{\beta \gamma }$ with respect to a reference system, for which both leaflets of the membrane are exposed to the spectator phase γ [28]. In what follows, the shorter term “adhesive strength” will be used as an abbreviation for “adhesion free energy per area”. The adhesive strength $W_{\alpha \gamma }$ is negative if the membrane prefers the α over the γ phase and positive otherwise. Likewise, $W_{\beta \gamma }$ is negative if the membrane prefers the β over the γ phase. Using these parameters, the adhesion free energy $E_{\mathrm{ad}}$ of the vesicle-droplet system becomes
(35)$$E_{\mathrm{ad}}=W_{\alpha \gamma }A_{\alpha \gamma }+W_{\beta \gamma }A_{\beta \gamma }=W_{\alpha \gamma }A+\left(W_{\beta \gamma }-W_{\alpha \gamma }\right)A_{\beta \gamma }\;.$$The term $W_{\alpha \gamma }A$ represents the adhesion free energy of the vesicle when it is completely immersed in the α phase and the term $\left(W_{\beta \gamma }-W_{\alpha \gamma }\right)A_{\beta \gamma }$ corresponds to the change in the adhesion free energy when the β droplet displaces the α phase.

In addition, we now have to distinguish the three pressures $P_{\alpha }$, $P_{\beta }$, and $P_{\gamma }$ within the three liquid phases α, β, and γ. The corresponding pressure terms have a slightly different form for the two wetting morphologies displayed in Figure 1. If the condensate droplet adheres to the vesicle membrane from the exterior solution as in Figure 1a, we have to include constraints on the droplet volume $V_{\beta }$ and on the vesicle volume $V_{\gamma }$, which leads to the pressure-dependent contribution [28] (exterior phase separation, Figure 1a)
(36)$$F_{P}^{\mathrm{ex}}\equiv \left(P_{\alpha }-P_{\gamma }\right)V_{\gamma }+\left(P_{\alpha }-P_{\beta }\right)V_{\beta }$$
to the shape functional of the vesicle-droplet system. If the two coexisting liquid phases α and β are formed within the vesicle as in Figure 1b, we have to include constraints on the two droplet volumes $V_{\alpha }$ and $V_{\beta }$, which implies that the pressure-dependent contribution to the shape functional now has the form [5] (interior phase separation, Figure 1b)
(37)$$F_{P}^{\mathrm{in}}\equiv \left(P_{\gamma }-P_{\alpha }\right)V_{\alpha }+\left(P_{\gamma }-P_{\beta }\right)V_{\beta }.$$The shape functional of the vesicle-droplet system is then given by [5,28]
(38)$$F^{\mu }=F_{P}^{\mu }+\Sigma A+E_{\;\mathrm{be}}+\Sigma_{\alpha \beta }A_{\alpha \beta }+E_{\mathrm{ad}}\;\mathrm{with}\;\mu =\mathrm{ex}\;\mathrm{or}\;\mu =\mathrm{in},\;$$
which consists of the pressure-dependent term $F_{P}^{\mu }$ as given by Equation (Equation 36) or Equation (Equation 37); the term ΣA, which controls the total membrane area *A* of the vesicle by the lateral stress Σ; the bending energy $E_{\;\mathrm{be}}$ in Equation (Equation 29); the interfacial free energy $\Sigma_{\alpha \beta }A_{\alpha \beta }$ of the αβ interface with area $A_{\alpha \beta }$; and the adhesion free energy $E_{\mathrm{ad}}$ in Equation (Equation 35). In general, the shape energy in Equation (Equation 31) contains an additional fluid-elastic term corresponding to the line free energy of the contact line, which will be ignored until Section 12 below.

### 8.5. Decomposition of Membrane Segment Tensions

The shape energy $F^{\mu }$ of the vesicle-droplet system as given by Equation (Equation 38) involves the Lagrange multiplier term ΣA, which controls the total membrane area *A* by the lateral stress Σ in the membrane. Using the decomposition of the membrane area, $A=A_{\alpha \gamma }+A_{\beta \gamma }$, this Lagrange multiplier term becomes equal to $\Sigma (A_{\alpha \gamma }+A_{\beta \gamma })$. When we combine this term with the adhesion free energy $E_{\mathrm{ad}}$ in Equation (Equation 35), we obtain
(39)$$\Sigma A+E_{\mathrm{ad}}=\Sigma_{\alpha \gamma }^{m}A_{\alpha \gamma }+\Sigma_{\beta \gamma }^{m}A_{\beta \gamma }$$
with the decomposition
(40)$$\Sigma_{\alpha \gamma }^{m}=\Sigma +W_{\alpha \gamma }\;\mathrm{and}\;\Sigma_{\beta \gamma }^{m}=\Sigma +W_{\beta \gamma }$$
for the tensions $\Sigma_{\alpha \gamma }^{m}$ and $\Sigma_{\beta \gamma }^{m}$ of the two membrane segments αγ and βγ.

Both segment tensions depend on the lateral stress Σ and, thus, on the size and shape of the vesicle. However, the affinity contrast *W*, which was defined in Equation (Equation 15), now becomes
(41)$$W=\Sigma_{\beta \gamma }^{m}-\Sigma_{\alpha \gamma }^{m}=W_{\beta \gamma }-W_{\alpha \gamma }$$
which is independent of the lateral stress Σ. Likewise, the rescaled affinity contrast *w* becomes
(42)$$w=\frac{\Sigma_{\beta \gamma }^{m}-\Sigma_{\alpha \gamma }^{m}}{\Sigma_{\alpha \beta }}=\frac{W_{\beta \gamma }-W_{\alpha \gamma }}{\Sigma_{\alpha \beta }}$$
which depends on the adhesion strengths $W_{\beta \gamma }$ and $W_{\alpha \gamma }$ as well as on the interfacial tension $\Sigma_{\alpha \beta }$ but not on the lateral stress Σ.

It is useful to view the terms $\Sigma_{\alpha \gamma }^{m}A_{\alpha \gamma }+\Sigma_{\beta \gamma }^{m}A_{\beta \gamma }$ in Equation (Equation 39), which are equal to the terms $\Sigma A+E_{\mathrm{ad}}$ of the shape functional in Equation (Equation 38), from a slightly different perspective. Instead of focusing on the total membrane area *A* as well as on the adhesive strengths $W_{\alpha \gamma }$ and $W_{\beta \gamma }$ of the two membrane segments, we may also focus on the two segment areas $A_{\alpha \gamma }$ and $A_{\beta \gamma }$ and interpret the two segment tensions $\Sigma_{\alpha \gamma }^{m}$ and $\Sigma_{\beta \gamma }^{m}$ as two Lagrange multipliers, which can be used to control the two segment areas. In the limit of low segment tension $\Sigma_{\alpha \gamma }^{m}$ corresponding to the corner point with x=0 and y=1 in Figure 11, the area $A_{\alpha \gamma }$ of the αγ segment is no longer constrained but can be changed to reduce the total energy of the vesicle-droplet system. Likewise, in the limit of low segment tension $\Sigma_{\beta \gamma }^{m}$ corresponding to the corner point with x=1 and y=0 in Figure 11, the membrane segment βγ can adapt its area $A_{\beta \gamma }$ to reduce the total energy of the system. For constant membrane area $A=A_{\alpha \gamma }+A_{\beta \gamma }$, changes in the segment areas $A_{\alpha \gamma }$ and $A_{\beta \gamma }$ imply a transfer of membrane area from one segment to the other.

### 8.6. Transfer of Membrane Area between Membrane Segments

When we transfer the membrane area ΔA from the αγ to the βγ membrane segment, we increase the area $A_{\beta \gamma }$ of the βγ segment by ΔA and decrease the area of the αγ segment by the same amount. The adhesion energy $E_{\mathrm{ad}}$ in Equation (Equation 35) is then changed from $E_{\mathrm{ad}}$ to $E_{\mathrm{ad}}+\Delta E_{\mathrm{ad}}$ with
(43)$$\Delta E_{\mathrm{ad}}=\left(W_{\beta \gamma }-W_{\alpha \gamma }\right)\Delta A=W\Delta A$$Both the affinity contrast *W* and the change in adhesion energy, $\Delta E_{\mathrm{ad}}$, are *negative* when the membrane prefers the β phase over the α phase, corresponding to partial wetting by the β phase. In such a situation, the membrane can gain adhesion energy by transfering some membrane area ΔA from the αγ to the βγ segment. As explained in the previous subsection, such an increase of the area $A_{\beta \gamma }$ is possible in the limit of low segment tension $\Sigma_{\beta \gamma }^{m}$ corresponding to the corner point with x=1 and y=0 in Figure 11. In this limit, the contact angle $\theta_{\alpha }$ approaches the value $\pi =180^{\circ }$ as follows from the tension triangle in Figure 10b. Such a behavior of the contact angle $\theta_{\alpha }$ together with a concomitant increase of the segment area $A_{\beta \gamma }$ has been recently observed for glycinin-rich condensate droplets adhering to GUV membranes. Ref. [61] The excess area ΔA was stored in membrane protrusions, which had the form of buds, fingers, or wave-like shape deformations.

## 9. Intrinsic Contact Angles at Smoothly Curved Membranes

### 9.1. “No Kink” Requirement and Smoothly Curved Membranes

The bending energy is an area integral over the (local) bending energy density as in Equation (Equation 29) which depends on the (local) mean curvature *M*. A kink in the membrane contour corresponds to the limit in which the curvature radius of the contour goes to zero. In this limit, the bending energy becomes infinite. This singular limit of the bending energy can be understood by looking at half a cylinder with curvature radius $R_{\mathrm{cy}}$ in the limit of small $R_{\mathrm{cy}}$. To avoid such unphysical behavior, we require that the membrane has no kinks and is smoothly curved along the contact line. This requirement reduces the three apparent contact angles to two intrinsic contact angles, $\theta_{\alpha }^{*}$ and $\theta_{\beta }^{*}$, as shown in Figure 15. Inspection of this figure reveals that these two contact angles now satisfy the relation
(44)$$\theta_{\alpha }^{*}+\theta_{\beta }^{*}=\pi =180^{\circ }$$
because the third contact angle $\theta_{\gamma }=\pi =180^{\circ }$. In Figure 15, the dashed black line represents the plane tangent to the membrane at the contact line. More precisely, this plane represents the common tangent to both membrane segments αγ and βγ at the contact line. Therefore, the term “smoothly curved” as used here implies that both membrane segments have the same tangent plane at the contact line.

### 9.2. Affinity Contrast from Intrinsic Contact Angles

Projecting the three surface tensions onto this common tangent plane, we obtain the tangential force balance as given by
(45)$$\Sigma_{\alpha \gamma }^{m}=\Sigma_{\beta \gamma }^{m}+\Sigma_{\alpha \beta }\cos\theta_{\beta }^{*}=\Sigma_{\beta \gamma }^{m}-\Sigma_{\alpha \beta }\cos\theta_{\alpha }^{*}$$
where the second equality follows from $\theta_{\beta }^{*}=\pi -\theta_{\alpha }^{*}$. Note that the tangential force balance in Equation (Equation 45) does not involve any curvature-elastic parameter such as the bending rigidity or the spontaneous curvature. Using the definition of the rescaled affinity contrast *w* in Equation (Equation 18), the tangential force balance now has the form
(46)$$w=\frac{\Sigma_{\beta \gamma }^{m}-\Sigma_{\alpha \gamma }^{m}}{\Sigma_{\alpha \beta }}=\cos\theta_{\alpha }^{*}=-\cos\theta_{\beta }^{*}$$
which provides a direct and simple relation between the affinity contrast *w* and the intrinsic contact angles $\theta_{\alpha }^{*}$ and $\theta_{\beta }^{*}$. Complete dewetting of the membrane from the β phase now corresponds to $\theta_{\alpha }^{*}=0$ and $\theta_{\beta }^{*}=\pi$, which implies the affinity contrast w=1. Likewise, complete wetting by the β phase is obtained for $\theta_{\beta }^{*}=0$ and $\theta_{\alpha }^{*}=\pi$, corresponding to w=−1. Furthermore, partial wetting by the β phase leads to −1<w<0, balanced adhesion to w=0, and partial wetting by the α phase is characterized by 0<w<1. Thus, the force balance regime in Figure 11 remains unchanged and has the same form as obtained from the analysis of the apparent contact angles $\theta_{\alpha }$, $\theta_{\beta }$, and $\theta_{\gamma }$.

The tangential force balance between the surface tensions as given by Equation (Equation 45) was first derived for axi-symmetric vesicle-droplet shapes by minimizing the combined bending and adhesion energy, making the simplifying assumption that both membrane segments αγ and βγ have zero spontaneous curvature [5]. The same tangential force balance also applies if both membrane segments have the same spontaneous curvature [28]. Thus, for membranes with uniform curvature-elastic parameters κ and *m*, the tangential force balance in Equation (Equation 45) does not depend on these curvature-elastic parameters. If the two membrane segments have different spontaneous curvatures, the tangential force balance involves additional terms which reflect discontinuities of the mean curvature along the contact line [28]. So far, such discontinuities have not been observed experimentally.

### 9.3. “No Kink” Requirement and Continuity of Mean Curvature

As mentioned, the tangential force balance as displayed in Figure 15 and described by Equation (Equation 45) follows from the requirement that the two membrane segments have a common tangent plane at the contact line. For an axisymmetric shape parametrized by the arc length *s* and the tilt angle ψ=ψ(s) of the normal vector [70], this requirement implies that ψ(s) is continuous across the circular contact line. It turns out that, for minimal energy shapes, continuity of ψ(s) leads to the more stringent condition that dψ/ds is also continuous at the contact line as first obtained for the analogous geometry of a circular domain boundary separating two intramembrane domains of the vesicle membrane [78].

The continuity of dψ/ds implies the continuity of the mean curvature *M* across the contact line. One should note that this boundary condition at the contact line between membrane and droplet is different from the corresponding boundary condition at the contact line between the membrane and a solid or rigid particle of radius $R_{\mathrm{pa}}$. If the particle adheres to the membrane from the exterior solution, the bound membrane segment, which is analogous to the βγ membrane segment in contact with the adhering droplet, has the mean curvature $-1/R_{\mathrm{pa}}$ whereas the mean curvature of the unbound membrane segment, which is analogous to the αγ segment, exhibits the contact mean curvature $M_{\mathrm{co}}=-1/R_{\mathrm{pa}}+1/R_{W}$ along the contact line where $R_{W}=\sqrt{2\kappa /|W|}$ is the so-called adhesion length [79,80]. Thus, in the case of an adhering solid particle, the mean curvature of the membrane is discontinuous and jumps along the contact line.

### 9.4. Relation between Apparent and Intrinsic Contact Angles

Because the rescaled affinity contrast *w* can be expressed both in terms of the apparent contact angles as in Equation (Equation 24) and in terms of the intrinsic contact angles as in Equation (Equation 46), a combination of these two equations leads to the relationship
(47)$$\cos\theta_{\alpha }^{*}=-\cos\theta_{\beta }^{*}=\frac{\sin\theta_{\alpha }-\sin\theta_{\beta }}{\sin\theta_{\gamma }}$$
between the apparent and the intrinsic contact angles. This relationship has been confirmed by two different experimental studies as described by Figure 16.

In the first experimental study, a batch of 63 GUVs has been prepared using the same lipid composition and the same solution conditions [5]. The resulting vesicle-droplet couples had different sizes and different shapes. In particular, the vesicles differed in their volume-to-area ratio *v* which is defined by
(48)$$v=\frac{V}{\frac{4\pi }{3}R_{\mathrm{ve}}^{3}}\;\mathrm{with}\;R_{\mathrm{ve}}\equiv \sqrt{A/(4\pi )}.$$This parameter has the limiting value v=1 for a spherical shape of the GUV and v<1 for any other vesicle shape. The apparent contact angles as observed for this batch of GUVs varied over a large range, see Figure 16a. However, when these apparent angles were inserted into Equation (Equation 47) to compute the intrinsic contact angle $\theta_{\alpha }^{*}$, the latter angle was found to be roughly constant as shown in Figure 16b.

In a second more recent experimental study, the intrinsic contact angle $\theta_{\alpha }^{*}$ was determined for several batches of GUVs that contained different polymer concentrations as controlled by the ratio between the osmolarity of the exterior aqueous solution and the initial osmolarity of the interior solution [52]. For each osmolarity ratio, the intrinsic contact angle $\theta_{\alpha }^{*}$ was determined by two different experimental procedures. First, this angle was directly measured by super-resolution STED microscopy, leading to the first set of data (half-filled circles) displayed in Figure 16c. In addition, the apparent contact angles were also measured for different osmolarity ratios, and the intrinsic angle $\theta_{\alpha }^{*}$ was again computed from these apparent contact angles via Equation (Equation 47), see the second set of data (open triangles) in Figure 16c.

### 9.5. Force Balance Perpendicular to the Membrane

So far, we focused on the tangential force balance between the three surface tensions as described by Equation (Equation 45). As mentioned, this tangential force balance can also be obtained by minimizing the combined bending and adhesion energy for axisymmetric vesicle-droplet morphologies, provided the membrane segments αγ and βγ have the same bending energy and the same spontaneous curvature. For these vesicle-droplet systems, one can also derive an explicit form for the normal component of the force balance. The axisymmetric shape can be parametrized in terms of the arc length *s* and the tilt angle ψ=ψ(s) of the normal vector which leads to the principal curvature $C_{1}=d\psi /ds$ of the shape contour [70,78]. The normal force balance then has the form [5]
(49)$$\frac{d^{2}\psi }{ds^{2}}\left(s_{\mathrm{co}}\right){|}_{\beta \gamma }-\frac{d^{2}\psi }{ds^{2}}\left(s_{\mathrm{co}}\right){|}_{\alpha \gamma }=\frac{\Sigma_{\alpha \beta }}{\kappa }\cos\left(\theta_{\alpha }^{*}\right)=-\frac{\Sigma_{\alpha \beta }}{\kappa }\cos\left(\theta_{\beta }^{*}\right)$$
which describes a jump in the derivative of the contour curvature dψ(s)/ds at the contact line with arc length $s=s_{\mathrm{co}}$.

The normal force balance in Equation (Equation 49) depends on the parameter combination $\Sigma_{\alpha \beta }/\kappa$ which involves the interfacial tension $\Sigma_{\alpha \beta }$ and the bending rigidity κ of the membrane. The inverse parameter combination, $\kappa /\Sigma_{\alpha \beta }$, has the dimension of a squared length. Dimensional analysis implies that $\sqrt{\kappa /\Sigma_{\alpha \beta }}$ sets the scale for the contour curvature radius, $1/C_{1}$, at the contact line. This conclusion is confirmed by a more detailed theoretical analysis that examines the shape of the highly curved membrane segments close to a contact line as observed by super-resolution STED microscopy, see Figure 16c. Note that the curvature radius $\sqrt{\kappa /\Sigma_{\alpha \beta }}$ becomes large for large bending rigidity κ but small for large interfacial tension $\Sigma_{\alpha \beta }$. Thus, the curvature radius $\sqrt{\kappa /\Sigma_{\alpha \beta }}$ encodes the competition between the bending resistance of the membrane and the capillary forces exerted by the interfacial tension onto the contact line.

### 9.6. Threshold of Droplet Size for the Onset of Adhesion

The length scale $\sqrt{\kappa /\Sigma_{\alpha \beta }}$, which enters the normal force balance in Equation (Equation 49) and determines the highly curved membrane segment along the contact line, is also important in order to understand the onset of adhesion. Thus, consider a spherical β droplet of radius $R_{\mathrm{dr}}$ in the vicinity of a vesicle membrane. Both droplet and vesicle are initially immersed in the liquid phase α as in Figure 2a. Furthermore, as long as the vesicle membrane is in contact with the α phase, it is subject to the membrane tension $\Sigma_{\alpha \gamma }^{m}$. When the droplet comes into contact with the vesicle membrane, it creates a small contact area, $\Delta A_{\beta \gamma }$, which experiences the membrane tension $\Sigma_{\beta \gamma }^{m}$. At the same time, both the area of the αβ interface and the area of the αγ membrane segment are reduced by $\Delta A_{\beta \gamma }$. As a consequence, the creation of the small contact area involves the adhesion energy
(50)$$\Delta E_{\mathrm{ad}}=-\Sigma_{\alpha \beta }\Delta A_{\beta \gamma }+\left(\Sigma_{\beta \gamma }^{m}-\Sigma_{\alpha \gamma }^{m}\right)\Delta A_{\beta \gamma }=\left(-1+w\right)\Sigma_{\alpha \beta }\Delta A_{\beta \gamma }$$
where the second equality follows from the definition of the rescaled affinity contrast *w* in Equation (Equation 18). Because the affinity contrast satisfies the inequalities −1≤w≤+1, the adhesion energy $E_{\mathrm{ad}}$ is negative unless the affinity contrast attains the limiting value w=+1, which describes complete dewetting of the membrane from the β droplet.

When the condensate droplet comes into contact with the vesicle membrane, it will impose its curvature $1/R_{\mathrm{dr}}$ onto the membrane. The membrane segment βγ with the small area $\Delta A_{\beta \gamma }$ will then acquire the bending energy
(51)$$\Delta E_{\;\mathrm{be}}=\frac{2\kappa }{R_{\mathrm{dr}}^{2}}\Delta A_{\beta \gamma }$$
as follows from Equation (Equation 29) when we ignore the spontaneous curvature *m*. The total energy change caused by the formation of the small contact area $\Delta A_{\beta \gamma }$ is then given by
(52)$$\Delta E=\Delta E_{\mathrm{ad}}+\Delta E_{\;\mathrm{be}}=\Delta A_{\beta \gamma }\left[\left(-1+w\right)\Sigma_{\alpha \beta }+\frac{2\kappa }{R_{\mathrm{dr}}^{2}}\right]$$
which must be negative to favor the adhesion of the droplet to the membrane. Thus, the droplet starts to adhere to the membrane for $\left(-1+w\right)\Sigma_{\alpha \beta }+2\kappa /R_{\mathrm{dr}}^{2}<0$ which implies that the droplet radius $R_{\mathrm{dr}}$ must exceed a certain threshold value $R_{\mathrm{dr}}^{o}$ as described by the inequality
(53)$$R_{\mathrm{dr}}>R_{\mathrm{dr}}^{o}\equiv \sqrt{\frac{\kappa }{\Sigma_{\alpha \gamma }}}\sqrt{\frac{2}{1-w}}$$
for the droplet radius. The threshold value $R_{\mathrm{dr}}^{o}$ for the droplet size attains its smallest value, which is equal to $\sqrt{\kappa /\Sigma_{\alpha \beta }}$, for rescaled affinity contrast w=−1, which corresponds to complete wetting of the membrane by the β droplet. In addition, this threshold radius grows as $1/\sqrt{1-w}$ when we approach the limiting value w=+1, corresponding to complete dewetting of the membrane from the β droplet.

The threshold $R_{\mathrm{dr}}^{o}$ as given by Equation (Equation 53) encodes the competition between bending rigidity κ, interfacial tension $\Sigma_{\alpha \beta }$, and rescaled affinity contrast *w* but ignores the possible influence of the spontaneous curvature of the membrane and the line tension of the contact line. A significant spontaneous curvature will affect this threshold, depending on the sign of this curvature. Indeed, when the droplet approaches the membrane from the exterior solution, corresponding to an endocytic process, a negative spontaneous curvature will facilitate the onset of adhesion whereas a positive spontaneous curvature will impede this onset, in analogy to the onset of adhesion for solid nanoparticles [80]. Because adhesion starts with a nanoscopic membrane segment, the line tension of the contact line will also affect the threshold value $R_{\mathrm{dr}}^{o}$. As described in Section 12 below, the line tension of the vesicle-droplet system can be positive or negative. A negative line tension acts to facilitate the onset of adhesion whereas a positive line tension acts to delay this onset.

## 10. Spontaneous Curvature and Formation of Membrane Nanotubes

The vesicle-droplet system can follow another morphological pathway when the vesicle membrane in contact with the condensate phase acquires a relatively large spontaneous curvature, which provides a quantitative measure for the transbilayer asymmetry of the membrane. Each biomembrane is built up from a lipid bilayer, which consists of two leaflets that can differ in their molecular composition and can be exposed to different aqueous solutions. These transbilayer asymmetries can generate a significant spontaneous (or preferred) curvature of the membrane. If this spontaneous curvature is large compared to the inverse size $1/R_{\mathrm{ve}}$ of the mother vesicle, the vesicle membrane forms membrane nanotubes as observed for the αγ membrane segment in contact with the PEG-rich α phase [7,8,9]. In addition to a large spontaneous curvature, the formation of nanotubes requires osmotic deflation of the vesicle volume in order to release some excess membrane area that can be stored in the nanotubes.

### 10.1. Transbilayer Asymmetry and Spontaneous Curvature

On the molecular scale, the transbilayer asymmetry of bilayer membranes can arise from many different mechanisms [48,81]. One such mechanism is provided by the adsorption of macromolecules onto the bilayers. For aqueous two-phase systems of PEG and dextran, the adsorption of PEG molecules was identified as the dominant mechanism for the transbilayer asymmetry of the membranes [9]. This conclusion was corroborated by atomistic molecular dynamics simulations. The lipid bilayers studied in the simulations and in the experiments had the same compositions of DOPC, DPPC, and cholesterol, forming a liquid-disordered (Ld) and a liquid-ordered (Lo) bilayer phase. Likewise, the simulated PEG chains had a length of 180 monomers, corresponding to the average molecular weight of the PEG studied in the experiments.

Snapshots of the molecular dynamics simulations as in Figure 17 revealed that the PEG molecules are only weakly bound to the lipid bilayer. The two terminal OH groups of each PEG molecule were frequently bound to the membrane via hydrogen bonds. In addition, a small number of contacts was formed between the polymer backbones and the membranes. Combining both types of contacts, the adsorbed polymers formed an average number of about 4.5 and 3.2 contacts with the liquid-ordered and the liquid-disordered membranes, respectively. A more quantitative measure for the affinity of the polymers to the membranes is provided by the potential of mean force. The computation of this potential indicated that the PEG molecules have essentially the same affinity for both types of membranes, with a binding free energy of about 4 kJ/mol or $1.6\;k_{B}T$ per polymer chain [9].

In the experimental studies, the two leaflets of the lipid bilayers were exposed to different PEG concentrations in the adjacent aqueous solutions which generated asymmetric adsorption layers and, thus, a significant transbilayer asymmetry. In fact, the corresponding spontaneous curvature was surprisingly large and led to the spontaneous formation of membrane nanotubes that protruded into the PEG-rich phase within the interior compartment of giant vesicles as described in the next subsection. Three different computational methods were used to determine the magnitude of this spontaneous curvature. As a result, the spontaneous curvature was estimated to be of the order of −1/(100 nm) for the lipid bilayers in the Ld phase and of the order of −1/(1000 nm) for those in the Lo bilayer phase. Note that the spontaneous curvature is negative which takes into account that the nanotubes protrude into the interior compartment of the vesicles, see Figure 18 and Figure 19. The negative sign of the spontaneous curvature as observed experimentally agrees with theoretical and computational studies [82,83] which predict that the membrane bulges towards the leaflet with the more densely packed adsorption layer.

### 10.2. Different Patterns of Membrane Nanotubes

The spontaneous tubulation of giant vesicles leads to three different patterns of nanotubes, depending on the polymer concentration inside the vesicles. This concentration can be controlled by the osmolarity of the exterior solution. For small exterior osmolarities and small interior polymer concentrations, the interior solution attains a spatially uniform liquid phase, corresponding to the one-phase region in the phase diagrams of Figure 6 and Figure 13. The giant vesicle then forms the tube pattern denoted by VM-A in Figure 18.

Crossing the binodal line of the phase diagram by increasing the exterior osmolarity and thus the interior polymer concentration, the interior polymer solution undergoes phase separation. Close to the critical point, the phase-separated polymer solution leads to the tube pattern VM-B in Figure 18, for which a confocal image is displayed in Figure 19a. This VM-B pattern is observed when the polymer concentrations of the interior solution belongs to the complete wetting (CW) subregion in the phase diagram of Figure 13a. In this case, the vesicle membrane is completely wetted by the PEG-rich α phase, which spatially separates the droplet of the dextran-rich β phase from the membrane, as displayed in Figure 13b without the nanotubes. These nanotubes explore the whole PEG-rich α phase but stay away from the dextran-rich β phase, see VM-B pattern in Figure 18.

Finally, yet another pattern of nanotubes is observed when the interior PEG-dextran solution belongs to the partial wetting (PW) subregion of the phase diagram in Figure 13a. For partial wetting by the α phase, the vesicle membrane is in contact with both the PEG-rich α and the dextran-rich β phase, as displayed in Figure 13c without the nanotubes. The nanotubes now adhere to the αβ interface between the α and β droplets and form the VM-C pattern in Figure 18. A confocal image of this pattern is shown in Figure 19b. In fact, the distinction between the VM-C and the VM-B patterns of membrane nanotubes provides a very useful method to distinguish partial from complete wetting because the location of the fluorescently labeled nanotubes can be directly observed by fluorescence microscopy, in contrast to the location of the αβ interface

The spontaneous tubulation of GUVs exposed to PEG-dextran two-phase systems was first observed and analyzed in Refs. [7,8]. Much denser and more complex tube patterns have been recently imaged by super-resolution STED microscopy [84]. The latter experiments also revealed that the nanotubes can undergo shape transformations into double-membrane sheets and that this transformation proceeds via a fascinating growth process, typically starting from the interior ends of the individual tubes.

### 10.3. Spontaneous Tubulation without Liquid-Liquid Phase Separation

The tubulation of the αγ membrane segments in contact with the PEG-rich α condensate is driven by the large spontaneous curvature of the αγ segments, arising from the different PEG concentrations in the interior and exterior solution which lead to a different density of PEG adsorbed onto the two leaflets of the bilayer membranes. Analogous tubulation processes are expected to occur for other vesicle membranes provided they have a sufficiently large spontaneous curvature [8]. This expectation has been confirmed for several vesicle systems.

One example for the spontaneous tubulation of GUV membranes, which are not in contact with aqueous two-phase systems, is provided by the VM-A pattern in Figure 18. This pattern of membrane nanotubes is formed when the vesicle membrane is exposed to two different but uniform liquid phases in the interior and exterior solution. Another example for such a tubulation process has been observed for GUVs that were exposed to PEG-sucrose solutions with a higher PEG concentration in the interior compared to the exterior solution. Some examples for tabulated vesicles in the absence of dextran are displayed in Figure 20. Third, spontaneous tubulation of giant vesicles has also been observed when the vesicle membranes contained the phospholipid POPC and a small amount of the glycolipid GM1, see Figure 21. In the latter case, the magnitude of the spontaneous curvature was about −1/(155 nm) and −1/(95 nm) for lipid bilayers prepared with 2 and 4 mol % GM1, respectively [85].

The glycolipid GM1 has attracted much recent interest because it is abundant in all mammalian neurons [86] and plays an important role in many neuronal processes and diseases [87]. Furthermore, GM1 acts as a membrane anchor for various toxins, bacteria, and viruses such as the simian virus 40 [88].

## 11. Engulfment of Condensate Droplets by Vesicle Membranes

In the previous section, we discussed the response of the vesicle-droplet system to osmotic deflation when one of the membrane segments has a large spontaneous curvature, which leads to the formation of membrane nanotubes. Now, we consider the response of the vesicle-droplet system when the morphological behavior is not governed by a large spontaneous curvature but rather by a large magnitude of the interfacial tension $\Sigma_{\alpha \beta }$. In order to reduce the free energy contribution $\Sigma_{\alpha \beta }A_{\alpha \beta }$ of the αβ interface, the vesicle membrane can engulf the droplet, thereby decreasing the interfacial area $A_{\alpha \beta }$. In the following, we first look at partial and complete engulfment of condensate droplets by giant vesicles and at partial engulfment by nanovesicles. We also consider stalled engulfment processes that arise when the membrane area is too small to completely engulf a large droplet.

### 11.1. Partial and Complete Engulfment by Giant Vesicles

When a condensate droplet adheres to the membrane of a giant vesicle, it can become partially or completely engulfed by the droplet as in Figure 3b,c, respectively. In Figure 22, these two microscopy images are compared with schematic drawings of the adhesion morphology. The transformation from partial to complete engulfment can again be controlled by osmotic deflation which leads to a reduction of the vesicle volume. During this transformation, the interfacial area $A_{\alpha \beta }$ of the αβ interface decreases, thereby decreasing the interfacial contribution $\Sigma_{\alpha \beta }A_{\alpha \beta }$ to the free energy of the vesicle-droplet system.The interfacial area $A_{\alpha \beta }$ vanishes for complete engulfment as in Figure 22c,d.

On the other hand, complete engulfment also increases the bending energy of the membrane, which is proportional to the bending rigidity κ. If we ignore the spontaneous curvature of the membrane, the bending energy of the two spherical segments in Figure 22c,d is equal to 16πκ. This bending energy is independent of the size of the droplet whereas the interfacial free energy is proportional to the droplet’s surface area. Therefore, complete engulfment will be energetically favored by the gain in interfacial free energy when the size of the droplet exceeds a certain threshold value, which is proportional to $\sqrt{\kappa /\Sigma_{\alpha \beta }}$. Energy minimization for axisymmetric vesicle-droplet morphologies has confirmed this conclusion and provided details about the dependence of the engulfment process on the surface tensions and on the intrinsic contact angle [89].

### 11.2. Partial Engulfment by Nanovesicles

Partial engulfment of small condensate droplets by the membranes of nanovesicles has been observed in molecular dynamics simulations [30]. One example is shown in Figure 23, which was obtained for solute mole fraction $\Phi_{S}=0.004$ and solubility ζ=25/70=0.36, corresponding to the two-phase coexistence region of the phase diagram in Figure 6. Initially, both the nanodroplet and the nanovesicle are fully immersed in the liquid phase α as shown in Figure 23a. When the droplet gets into contact with the vesicle membrane, a small contact area is formed as in Figure 23b. After this onset of adhesion, the vesicle membrane starts to engulf the membrane. This process continues by pulling out membrane area from the thermally excited undulations, thereby increasing the lateral stress in the membrane. Eventually, a new stable morphology, corresponding to partial engulfment, is reached as shown in Figure 23c.

A further reduction of the vesicle volume for the partial engulfment morphology in Figure 23c will increase the contact area of the membrane segment βγ between droplet and membrane. This volume reduction process can lead to complete engulfment or to stalled engulfment, depending on the relative size of droplet and vesicle.

### 11.3. Stalled Engulfment for Sufficiently Large Droplets

When the condensate droplet exceeds a certain size compared to the linear dimension of the vesicle membrane, the deflation-induced engulfment process is stalled. To derive the corresponding threshold value for the droplet size, we start from the isoperimetric inequality
(54)$$A^{3}\geq 36\pi V^{2}\;,$$
which is valid for any closed surface with surface area *A* and enclosed volume *V* [90,91]. The limiting case $A^{3}=36\pi V^{2}$ applies to a spherical shape, which is the shape with the smallest possible surface area *A* for a given volume *V*.

We now apply the isoperimetric inequality to the vesicle-droplet morphology of complete engulfment as displayed in Figure 22c,d. For such a morphology, the vesicle membrane consists of two membrane segments, αγ and βγ, which have the surface areas $A_{\alpha \gamma }$ and $A_{\beta \gamma }$ and are connected by a narrow or closed membrane neck. The membrane area of this neck can be ignored compared to the segment areas $A_{\alpha \gamma }$ and $A_{\beta \gamma }$. The βγ segment provides the contact area with the droplet which has the volume $V_{\beta }$. Furthermore, the αγ segment with surface area $A_{\alpha \gamma }$ encloses the combined volume $V_{\beta }+V_{\gamma }$, where $V_{\gamma }$ is the volume of the γ phase. When we apply the isoperimetric inequality to this geometry, we obtain
(55)$$A_{\beta \gamma }\geq {\left(36\pi \right)}^{1/3}V_{\beta }^{2/3}\;\mathrm{and}\;A_{\alpha \gamma }\geq {\left(36\pi \right)}^{1/3}{\left(V_{\beta }+V_{\gamma }\right)}^{2/3}\;.$$Combining these two inequalities, the total membrane area $A=A_{\alpha \gamma }+A_{\beta \gamma }$ satisfies
(56)$$A\geq {\left(36\pi \right)}^{1/3}\left[{\left(V_{\beta }+V_{\gamma }\right)}^{2/3}+V_{\beta }^{2/3}\right]\geq 2{\left(36\pi \right)}^{1/3}V_{\beta }^{2/3}$$
where the second inequality follows from the inequality $V_{\gamma }\geq 0$ for the volume of the γ phase. The limiting case with $V_{\gamma }=0$ corresponds to two nested membrane segments which touch each other. Thus, complete engulfment is only possible if the total membrane area *A* exceeds the threshold value in Equation (Equation 56). Rewriting the latter equation, we also conclude that complete engulfment of the β droplet is only possible for a sufficiently small droplet volume $V_{\beta }$ that satisfies
(57)$$V_{\beta }\leq \frac{A^{3/2}}{2^{3/2}{\left(36\pi \right)}^{1/2}}$$
but impossible for droplet volumes that exceed this threshold value.

The process of stalled engulfment has been observed in molecular dynamics simulations as shown in Figure 24 [30]. In Figure 24a, we see a stalled engulfment process that proceeds in an axisymmetric manner as can be concluded from the circular shape of the contact line between the αβ interface and the vesicle membrane. in Figure 24b, the contact line starts with a circular shape but then undergoes a symmetry-breaking transition to a strongly noncircular shape. These different morphological pathways depend on the different numbers of lipids assembled in the two leaflets of the bilayer membranes and on the corresponding leaflet tensions [30].

## 12. Line Tension of Contact Line

In order to understand the axisymmetric and non-axisymmetric shapes of the contact lines in Figure 24a,b, we need to take another quantity into account which is provided by the line tension λ of the contact line. This line tension, which has the physical units of energy per length, can be positive or negative and becomes important for sufficiently small contact lines with a size that is comparable to or smaller than $\lambda /\Sigma_{\alpha \beta }$. This length scale encodes the competition between the line tension λ and the interfacial tension $\Sigma_{\alpha \beta }$ as follows from dimensional analysis and can be systematically derived from the force balance between the three surface tensions and the line tension.

### 12.1. Positive and Negative Line Tensions

The axisymmetric and non-axisymmetric vesicle-droplet systems arising from stalled engulfment as in Figure 24a,b are distinguished by the sign of the contact line tension. A positive line tension favors a circular shape of the contact line and the αβ interface whereas a negative line tension favors a non-circular shape of contact line and αβ interface as shown in Figure 24b. The contribution of the contact line to the free energy of the vesicle-droplet system is equal to $\lambda L_{\mathrm{co}}$, with the line tension λ and the length $L_{\mathrm{co}}$ of the contact line. A negative line tension implies that this free energy contribution is negative as well and that the contact line would like to increase its length $L_{\mathrm{co}}$. At the same time, the system would also like to reduce the area of the αβ interface, which is bounded by the contact line. Thus, the system tries to maximize the length of the contact line and to simultaneously minimize the area of the αβ interface. Both requirements can be satisfied by a non-circular, elongated shape of the contact line as in Figure 24b.

For liquid mixtures without lipid membranes, the notion of line tension was already introduced by Gibbs who called it ‘linear tension’ and pointed out that this tension may be positive or negative [54,92]. In contrast, interfacial tensions must always be positive as required by the thermodynamic stability of the interfaces. In the absence of membranes, negative values of the line tension have been observed for sessile liquid droplets on solid surfaces [93], for lense-shaped droplets between two bulk liquids [94], and in simulations of Lennard-Jones fluids [95]. Negative line tensions have also been found for Plateau borders in foams [96].

### 12.2. Interfacial Tension Versus Line Tension

As previously mentioned, the free energy contribution $E_{\mathrm{co}}$ arising from the contact line is given by $E_{\mathrm{co}}=\lambda L_{\mathrm{co}}$, where $L_{\mathrm{co}}$ denotes the length of the contact line. For comparison, the free energy contribution $E_{\alpha \beta }$ arising from the αβ interface is equal to $E_{\alpha \beta }=\Sigma_{\alpha \beta }A_{\alpha \beta }$ where $A_{\alpha \beta }$ is the area of the αβ interface. If the length $L_{\mathrm{co}}$ of the contact line is comparable to the linear dimensions of the αβ interface and, thus, to $\sqrt{A_{\alpha \beta }}$, the ratio of the line free energy to the interfacial free energy is given by
(58)$$\frac{E_{\mathrm{co}}}{E_{\alpha \beta }}∼\frac{\lambda \sqrt{A_{\alpha \beta }}}{\Sigma_{\alpha \beta }A_{\alpha \beta }}=\frac{\lambda }{\Sigma_{\alpha \beta }\sqrt{A_{\alpha \beta }}}∼\frac{\lambda }{\Sigma_{\alpha \beta }L_{\mathrm{co}}}$$
which decays as $1/\sqrt{A_{\alpha \beta }}∼1/L_{\mathrm{co}}$ for large interfacial area $A_{\alpha \beta }∼L_{\mathrm{co}}^{2}$. More precisely, the line tension contribution to the free energy becomes negligible when the contact line length $L_{\mathrm{co}}$ is large compared to $\lambda /\Sigma_{\alpha \beta }$. On the other hand, the line tension contribution will become important when the contact line length $L_{\mathrm{co}}$ becomes comparable to or smaller than $\lambda /\Sigma_{\alpha \beta }$.

### 12.3. Force Balance between Surface Tensions and Line Tension

The line tension contributes the additional free energy term $E_{\mathrm{co}}=\lambda L_{\mathrm{co}}$ to the shape functional of the vesicle-droplet system as given by Equation (Equation 38). Minimization of this shape functional then leads to a force balance relation between the three surface tensions and the line tension λ. For axisymmetric shapes which can be parameterized by the arc length of the shape contour, the contact line is circular with radius $R_{\mathrm{co}}$ and located at a certain arc length $s=s_{\mathrm{co}}$, where the normal vector is tilted by the angle $\psi_{\mathrm{co}}=\psi \left(s_{\mathrm{co}}\right)$. The tangential (or parallel) force balance is then given by [5]
(59)$$\Sigma_{\beta \gamma }^{m}-\Sigma_{\alpha \gamma }^{m}=\Sigma_{\alpha \beta }\cos\theta_{\alpha }^{*}+\frac{\lambda }{R_{\mathrm{co}}}\cos\psi_{\mathrm{co}}$$
which is equivalent to the affinity contract
(60)$$W=\frac{\Sigma_{\beta \gamma }^{m}-\Sigma_{\alpha \gamma }^{m}}{\Sigma_{\alpha \beta }}=\cos\theta_{\alpha }^{*}+\frac{\lambda }{\Sigma_{\alpha \beta }R_{\mathrm{co}}}\cos\psi \left(s_{\mathrm{co}}\right)\;.$$Both the tangential force balance and the affinity contrast now involve correction terms proportional to the line tension λ and inversely proportional to the radius $R_{\mathrm{co}}$ of the contact line. Likewise, the line tension λ also affects the normal (or perpendicular) force balance at the contact line which now has the form [5]
(61)$$\frac{d^{2}\psi }{ds^{2}}\left(s_{\mathrm{co}}\right){|}_{\beta \gamma }-\frac{d^{2}\psi }{ds^{2}}\left(s_{\mathrm{co}}\right){|}_{\alpha \gamma }=\frac{\Sigma_{\alpha \beta }}{\kappa }\cos\left(\theta_{\alpha }^{*}\right)+\frac{\lambda }{\kappa R_{\mathrm{co}}}\sin\psi \left(s_{\mathrm{co}}\right)\;.$$

The λ-dependent terms in Equations (Equation 59)–(Equation 61) are significant when the radius $R_{\mathrm{co}}$ of the circular contact line is sufficiently small and satisfies
(62)$$R_{\mathrm{co}}≲\frac{\lambda }{\Sigma_{\alpha \beta }}\;.$$This condition is eventually fulfilled when the contact line and the adjacent membrane neck become closed during complete engulfment. As a consequence, the positive or negative sign of the line tension strongly affects the closure of the membrane neck.

## 13. Different Shapes of Closed Membrane Necks

### 13.1. Tight-Lipped Membrane Necks for Planar Bilayers

Negative values of the contact line tension were first observed in molecular dynamics simulations of condensate droplets adhering to planar lipid bilayers. The partial engulfment of such a droplet is displayed in Figure 25 as obtained for solute mole fraction $\Phi_{S}=0.0126$ and solubility ζ=1/2 in the phase diagram of Figure 6. The planar bilayer in Figure 25 is symmetric in the sense that each leaflet contains the same number of lipid molecules. Furthermore, this bilayer is subject to periodic boundary conditions, which can be used to control the mechanical tension within the bilayer.

In Figure 25, the bilayer experienced a significant bilayer tension that prevents this bilayer membrane from spreading over the whole droplet, as required for complete engulfment. Such an engulfment process was obtained as soon as the bilayer tension was reduced by decreasing the lateral size $L_{\|}$ of the simulation box; see Figure 26. This reduction of $L_{\|}$ was performed for a fixed number of lipid molecules within the bilayer and for constant volume $L_{\|}^{2}L_{z}$ of the simulation box. Because of the latter constraint, the reduction of $L_{\|}$ leads to an increase in the perpendicular box size $L_{z}$, as indicated in Figure 26.

For the planar and symmetric bilayers studied in [29], the reduction of the bilayer tension led to a tight-lipped membrane neck for a large range of interaction parameters. In order to form a tight-lipped neck, the line tension of the contact line must be negative. In general, negative line tensions lead to non-axisymmetric shapes of the contact line as in Figure 24b. Furthermore, such an elongated shape of the membrane neck prevents the fission of this neck, which is necessary for endocytosis of condensate droplets. Therefore, such an endocytic process has not been observed in the simulations of planar and symmetric bilayers. In contrast, nanovesicles were observed to undergo endocytosis and uptake of condensate droplets, provided the bilayers of these vesicles experienced a sufficiently large stress asymmetry between the two leaflets as explained in the next subsection [30].

The formation of a tight-lipped membrane neck implies an increase in the bending energy of the vesicle membrane [29]. Therefore, this unusual neck shape will be suppressed by a sufficiently large bending rigidity. The interplay between interfacial tension, bending rigidity, and negative line tension has also been studied by minimizing the combined bending and adhesion energy of the vesicle-droplet system [97]. The minimization was performed using the Surface Evolver algorithm [98] which is based on a triangulation of the membrane surface and is difficult to apply when the membrane shape involves narrow or closed membrane necks. On the other hand, the Surface Evolver calculations showed that the adhesion of a single condensate droplet can transform an axisymmetric vesicle into a non-axisymmetric vesicle-droplet morphology.

### 13.2. Stress Asymmetry between Two Bilayer Leaflets

The bilayer tensions $\Sigma_{\alpha \gamma }^{m}$ and $\Sigma_{\beta \gamma }^{m}$ of the two membrane segments αγ and βγ were essential in order to classify the adhesion geometries in terms of contact angles and to define the affinity contrast which provides a global view of the possible wetting transitions, see Figure 12. In order to obtain additional insight into complete engulfment and endocytosis of droplets, we will now consider the individual leaflets of the bilayers and decompose the bilayer tensions into two leaflet tensions. In the simulations, this decomposition of the bilayer tension can be obtained by partitioning the stress profile of the bilayer into two partial stress profiles associated with the two bilayer leaflets [83,99,100]. Each bilayer tension, $\Sigma_{\mathrm{bil}}$, is then decomposed according to
(63)$$\Sigma_{\mathrm{bil}}=\Sigma_{l1}+\Sigma_{l2}$$
where $\Sigma_{l1}$ and $\Sigma_{l2}$ represent the two leaflet tensions. In practice, this decomposition of the bilayer tension is feasible for planar bilayers [83,99] and for the bilayers of spherical nanovesicles [30,100] before these bilayers are deformed by an adhering droplet, see Figure 23a. All leaflet tensions discussed in the following represent such initial leaflet tensions of the undeformed bilayers.

It is important to realize that the two leaflets of a tensionless bilayer with $\Sigma_{\mathrm{bil}}=0$ typically experience significant leaflet tensions $\Sigma_{l1}$ and $\Sigma_{l2}$. Indeed, because of the decomposition $\Sigma_{\mathrm{bil}}=\Sigma_{l1}+\Sigma_{l2}$, the leaflet tensions of a tensionless bilayer satisfy $\Sigma_{l2}=-\Sigma_{l1}$. Therefore, for $\Sigma_{\mathrm{bil}}=0$, one leaflet tension is positive whereas the other leaflet tension is negative, corresponding to one stretched and one compressed leaflet, respectively.

In what follows, we will characterize the tensionless and undeformed bilayers by their initial stress asymmetry
(64)$$\Delta \Sigma =\Sigma_{l1}-\Sigma_{l2}=2\Sigma_{l1}=-2\Sigma_{l2}\;(\mathrm{for}\;\Sigma_{\mathrm{bil}}=0).$$This initial stress asymmetry is positive if the leaflet l1 is stretched and the leaflet l2 is compressed but negative if l1 is compressed and l2 is stretched. In the simulations, the initial stress asymmetry ΔΣ can be controlled by the lipid numbers that are assembled into the two bilayer leaflets. In addition, the initial stress asymmetry determines the shape of the membrane neck that is formed during complete engulfment of a condensate droplet.

### 13.3. Tight-Lipped Membrane Necks for Small Stress Asymmetries

For planar and symmetric bilayers, the two leaflets have identical leaflet tensions, $\Sigma_{l2}=\Sigma_{l1}$, which implies that the initial stress asymmetry ΔΣ is close to zero. As shown in Figure 26, such a bilayer forms a tight-lipped neck during the complete engulfment of a condensate droplet. The latter type of membrane neck was also observed for nanovesicle bilayers with a relatively small stress asymmetry [30]. One example is provided by a nanovesicle with $N_{ol}=5700$ and $N_{il}=4400$ as displayed in Figure 24b. When the bilayer tension of this nanovesicle is close to zero, the vesicle has the initial stress asymmetry $\Delta \Sigma =\Sigma_{ol}-\Sigma_{il}≃1.7\;k_{B}T/d^{2}$ between the leaflet tensions $\Sigma_{ol}$ and $\Sigma_{il}$ of the outer and inner leaflet where d≃0.8 nm is the bead diameter of the coarse-grained molecular model studied in the simulations.

The positive value of the initial stress asymmetry ΔΣ implies that the outer leaflet is stretched whereas the inner leaflet is compressed. In order to reduce this stress asymmetry, the bilayer prefers to bulge towards the inner leaflet, thereby increasing the area of the inner leaflet and decreasing the area of the outer one. When a β droplet with a diameter of 14d or 11.2 nm adheres to this vesicle, the droplet is completely engulfed by the vesicle membrane but the resulting contact line has the *negative* line tension $\lambda ≃-10\;k_{B}T/d$, which leads to a tight-lipped membrane neck during complete engulfment as observed in the simulations [30].

### 13.4. Axisymmetric Necks and Endocytosis for Large Stress Asymmetries

For sufficiently large stress asymmetries $\Delta \Sigma =\Sigma_{ol}-\Sigma_{il}>0$, the line tension λ of the contact line acquires a positive value [30]. One example is provided by a nanovesicle with $N_{ol}=5500$ lipids in its outer leaflet and $N_{il}=4600$ lipids in its inner leaflet. When the bilayer tension of the latter nanovesicle is close to zero, the vesicle has the initial stress asymmetry $\Delta \Sigma ≃2.7\;k_{B}T/d^{2}$. Adhesion of a droplet with a diameter of 11.2 nm then leads to a contact line with *positive* line tension $\lambda ≃+7k_{B}T/d$ and to a membrane neck that closes in an axisymmetric manner during complete engulfment, as shown in the first two snapshots of Figure 27. After the neck has been closed, it undergoes fission, thereby generating two nested daughter vesicles as in the last snapshot of Figure 27.

The transbilayer stress asymmetry plays the same role for nanovesicles as the spontaneous curvature for giant vesicles. In the latter case, the theory of curvature elasticity predicts that a sufficiently large spontaneous curvature generates a strong constriction force at the membrane neck that is sufficient to cleave the neck [101] as has been observed experimentally for giant unilamellar vesicles [73]. The endocytic process displayed in Figure 27 demonstrates an analogous fission mechanism for nanovesicles, with neck cleavage and vesicle division being induced by a sufficiently large transbilayer stress asymmetry.

## 14. Summary and Outlook

In this paper, recent results on membrane remodeling by the adhesion of condensate droplets have been reviewed and explained within the framework of fluid elasticity. The different adhesion morphologies were first discussed in a qualitative manner (Figure 1, Figure 2 and Figure 3) and then characterized in terms of the three apparent contact angles $\theta_{\alpha },\theta_{\beta }$, and $\theta_{\gamma }$, which can be measured by conventional fluorescence microscopy (Figure 8 and Figure 9). These contact angles are intimately related to the three surface tensions $\Sigma_{\alpha \beta }$, $\Sigma_{\alpha \gamma }^{m}$, and $\Sigma_{\beta \gamma }^{m}$ that act within the αβ interface as well as within the two membrane segments αγ and βγ. The three surface tensions balance each other along the contact line (Figure 10) and define the affinity contrast *W* between the membrane and the two liquid phases α and β as defined by Equation (Equation 15). The tensions $\Sigma_{\alpha \gamma }^{m}$ and $\Sigma_{\beta \gamma }^{m}$ of the membrane segments can be decomposed into a lateral stress Σ that is conjugate to the total membrane area and into the adhesion free energies per unit area, $W_{\alpha \gamma }$ and $W_{\beta \gamma }$, of the α and β phases at the membrane, see Equation (Equation 40).

The rescaled tensions $\Sigma_{\alpha \gamma }^{m}/\Sigma_{\alpha \beta }$ and $\Sigma_{\beta \gamma }^{m}/\Sigma_{\alpha \beta }$ of the two membrane segments as well as the rescaled affinity contrast $w=W/\Sigma_{\alpha \beta }$ can be directly expressed in terms of the apparent contact angles via Equations (Equation 22) and (Equation 24). Therefore, the rescaled affinity contrast *w*, which is a mechanical quantity, can be obtained by measuring the apparent contact angles, which are purely geometric quantities. On the other hand, the dimensionful affinity contrast $W=w\Sigma_{\alpha \beta }$ depends on the interfacial tension $\Sigma_{\alpha \beta }$ as well. For PEG-dextran solutions, the interfacial tension $\Sigma_{\alpha \beta }$ has been measured for a large part of the two-phase coexistence region (Figure 5). The rescaled affinity contrast allows us to obtain a global view of the force balance regime (Figure 11) where the vesicle-droplet morphology exhibits a stable contact line with balanced surface tensions. Approaching the boundaries of this force balance regime leads to partial-to-complete wetting transitions of the α and the β phases and to the complete engulfment of α and β droplets (Figure 12). Furthermore, the force balance regime includes two corner points, one of which plays a prominent role in a recent experimental study of glycinin-rich condensate droplets [61].

When we take into account that the vesicle membrane has a finite bending rigidity κ, the membrane should be smoothly curved along the contact line, as recently confirmed by super-resolution STED microscopy (Figure 14). Such a smoothly curved membrane implies that the three apparent contact angles are replaced by two intrinsic ones (Figure 15) which have been measured using two different experimental procedures (Figure 16). The bending rigidity does not affect the affinity contrast *W*, which is still defined by Equation (Equation 15), but the reduced affinity contrast *w* can now be expressed in terms of the intrinsic contact angles, see Equation (Equation 46). The competition between the interfacial tension $\Sigma_{\alpha \beta }$, which exerts capillary forces onto the membrane, and the bending rigidity κ, which acts to flatten the membrane, is encoded in the length scale $\sqrt{\kappa /\Sigma_{\alpha \beta }}$. This length enters the normal force balance as given by Equation (Equation 49), sets the scale for the small curvature radius of the membrane close to the contact line, and determines the threshold value for the onset of adhesion, see Equation (Equation 53). The latter relation ignores the membrane’s spontaneous curvature and the contact line tension, both of which are expected to affect the onset of adhesion but the influence of these two fluid-elastic parameters remains to be examined in a quantitative manner.

Condensate droplets can generate a large spontaneous curvature in the adjacent membrane segment, which leads to the spontaneous tubulation of this segment when we reduce the vesicle volume by osmotic deflation. Such a spontaneous tubulation process has been observed for giant vesicles exposed to phase-separated PEG-dextran solutions (Figure 18 and Figure 19). The diameter of the nanotubes is comparable to the inverse spontaneous curvature. For liquid-disordered vesicle membranes in contact with the PEG-rich phase, the nanotubes had a width of about 100 nm. Another fascinating remodeling process is provided by complete engulfment and endocytosis of condensate droplets. The latter process is strongly affected by the line tension of the contact line which can be positive or negative. For droplets adhering to planar bilayers, the line tension is typically negative and can then lead to an unusual tight-lipped membrane neck that suppresses membrane fission and droplet endocytosis (Figure 25 and Figure 26).

Molecular dynamics simulations of nanovesicles revealed that the sign of the line tension is determined by the stress asymmetry between the two leaflets of the bilayer [30]. The line tension is negative for relatively small stress asymmetries but positive for relatively large asymmetries. In the latter case, the membrane neck remains axisymmetric until the droplet is completely engulfed and then undergoes endocytosis, leading to the formation of two nested daughter vesicles, with the intraluminal vesicle enclosing the condensate droplet (Figure 27). For the necks of giant vesicles, we do not yet have experimental data, by which we could distinguish axisymmetric from non-axisymmetric neck shapes. Such a distinction should be accessible to super-resolution microscopy such as STED which provides a challenge for future experiments.

For phase-separated PEG-dextran solutions within giant vesicles as in Figure 1b, the formation of two daughter vesicles has also been observed. One daughter vesicle contained the PEG-rich α droplet whereas the other daughter vesicle was filled with the dextran-rich β droplet, but these two vesicles remained connected by a membrane nanotube (or tether) [10,102]. One possible explanation is that the latter systems had a negative line tension which would lead to a tight-lipped membrane neck, thereby impeding the fission of this neck. On the other hand, the connecting nanotube was observed to be quite long, with an extension of many micrometers, which raises the question about the location of the small, remaining αβ interface between the two coexisting aqueous phases. In order to clarify this issue experimentally, it should be useful to increase the spontaneous curvature of the giant vesicle membrane by binding His-tagged proteins to its outer leaflet, a process that leads to membrane fission even in the absence of aqueous phase separation [73].

In the molecular dynamics simulations of nanovesicles exposed to a binary mixture, intriguing morphological changes have also been observed in the one-phase region of this mixture when it was sufficiently close to the binodal line [50]. The vesicles formed prolate shapes in the absence of solute, corresponding to $\Phi_{S}=0$ and ζ=25/40=0.625 in the phase diagram of Figure 6. When solute was added to the exterior solution, it adsorbed onto the vesicle membrane and transformed the prolate into a dumbbell shape. For mole fraction $\Phi_{S}=0.025$ close to the binodal line, the dumbbell underwent recurrent shape transformations between dumbbells with closed and open necks. For $\Phi_{S}=0.026$ which is even closer to the binodal, the nanovesicle was divided up into two daughter vesicles, which continued to adhere to each other via an intermediate layer of adsorbed solutes. This solute-mediated adhesion turned out to be rather strong and difficult to overcome by changing the vesicle volume and/or the solute concentration. In fact, preliminary simulations indicate (Rikhia Ghosh, private communication) that such changes may induce fusion of the adhering daughter vesicles, thereby reversing the fission process. The relation between these fission processes observed in the one-phase region close to the binodal line [50] and those described here in the two-phase region (Figure 27) remains to be clarified.

The division of nanovesicles that form inward-pointing buds with exterior necks as shown in Figure 27 as well as the observed division of giant vesicles that form outward-pointing buds [73] involve only small changes in the vesicle shapes and therefore only small changes in the bending energies of their membranes. It then follows that, during neck fission, the main contribution to the free energy difference between the two daughter vesicles and the initial mother vesicle is provided by a change $\Delta E_{G}$ in the Gaussian curvature energy as given by Equation (Equation 28). During fission, the Euler characteristic χ is increased by Δχ=2 which leads to $\Delta E_{G}=2\pi \Delta \chi \kappa_{G}=4\pi \kappa_{G}$. Furthermore, the neck fission of a nanovesicle or a giant vesicle represents a spontaneous or exergonic process, that moves downhill in the free energy landscape, which implies $\Delta E_{G}<0$. Therefore, the Gaussian curvature modulus $\kappa_{G}$ must be negative, both for the endocytosis of condensate droplets by nanovesicles [30] and for the curvature-induced division of giant vesicles [73], in agreement with previous conclusions about this modulus [103,104,105,106].

Membrane fusion leads to the change Δχ=−2 of the Euler characteristic and to the change $\Delta E_{G}=-4\pi \kappa_{G}>0$ of the Gaussian curvature energy. For a negative Gaussian curvature modulus $\kappa_{G}<0$, the fusion process represents an uphill or endergonic process that is unlikely to occur unless it is coupled to another downhill or exergonic process. One downhill process that drives membrane fusion is the relaxation of membrane tension as observed in molecular dynamics simulations [107,108]. Membrane tension facilitates lipid flip-flops between two adhering membranes as well as the formation and opening of a fusion pore. In these simulation studies, the membrane fusion was induced by increasing the bilayer tension, $\Sigma_{\mathrm{bil}}=\Sigma_{l1}+\Sigma_{l2}$, without looking at the behavior of the individual leaflet tensions $\Sigma_{l1}$ and $\Sigma_{l2}$. It is conceivable that even tensionless bilayers with $\Sigma_{\mathrm{bil}}=0$ can fuse provided their leaflet tensions have a sufficiently large magnitude but this putative pathway remains a challenge for future simulations.

Another aspect of membrane fusion, that is closely related to the topic of this review, is the possibility that the fusion of two condensate droplets that adhere to different membranes leads to the fusion of these membranes. After the fusion of the two droplets, a condensate bridge will be formed between the two membranes which then experience capillary forces that can pull the membranes closer together. Any process that increases the interfacial tension of the capillary bridge will also increase the capillary force between the membranes. The bridging process can be modulated by the formation of intramembrane lipid domains which act to localize the capillary bridge within the domains and the capillary forces to the domain boundaries.

## Figures and Tables

**Figure 1 membranes-13-00223-f001:**
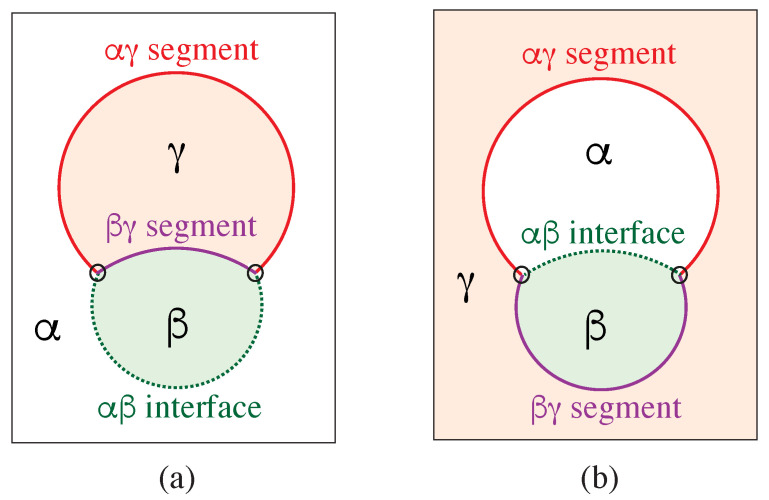
Geometry of vesicle-droplet systems which involve three liquid phases α (white), β (green), and γ (pink). The phases α and β represent two coexisting phases that arise via segregative or associative liquid–liquid phase separation: (**a**) Phase separation in the exterior solution and adhesion of the condensate droplet β to the outer leaflet of the vesicle membrane; and (**b**) Phase separation in the interior solution and adhesion of the β droplet to the interior leaflet of the membrane. The γ phase corresponds to an inert spectator phase. The αβ interface (dashed green line) and the vesicle membrane form the contact line (open circles) which partitions the vesicle membrane into two segments, the αγ segment exposed to the α and γ phases as well as the βγ segment in contact with the β and γ phases.

**Figure 2 membranes-13-00223-f002:**
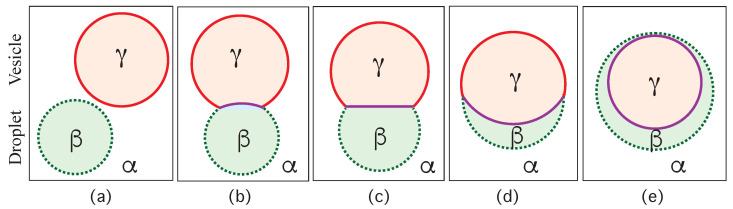
Different adhesion morphologies of a lipid vesicle (light red) interacting with a condensate droplet β (light green) that coexists with the bulk phase α (white): (**a**) Complete dewetting and (**b**) partial dewetting of the vesicle membrane (red/purple) by the condensate droplet β; (**c**) Balanced pressure between the β and the γ phase; (**c**) Partial wetting and (**d**) complete wetting of vesicle membrane by β droplet. As in Figure 1, all morphologies involve three aqueous phases, the liquid bulk phase α (white), the condensate phase β forming the droplet (light green), and the inert spectator phase γ (light red) within the vesicle. The contact area between droplet and membrane, which is equal to the surface area of the βγ segment (purple), increases from zero in (**a**) to the total membrane area in (**e**).

**Figure 3 membranes-13-00223-f003:**
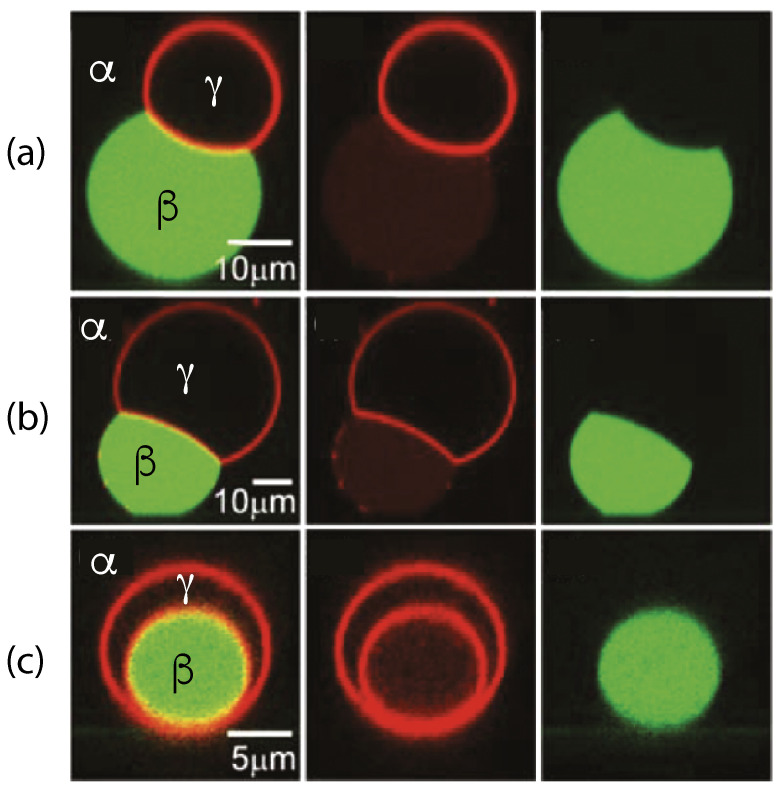
Adhesion morphologies of giant unilamellar vesicles (GUVs) exposed to exterior PEG-dextran solutions that undergo liquid–liquid phase separation into the PEG-rich bulk phase α (black) and the dextran-rich condensate droplet β (green): (**a**) Partial wetting of vesicle membrane by condensate droplet; (**b**) Partial wetting of the membrane by the droplet and partial engulfment of the droplet by the membrane; and (**c**) Complete engulfment of the droplet by the membrane which forms two spherical segments (red) connected by a narrow membrane neck, which is too small to be resolved. The middle column displays the red membrane channel, the right column the green droplet channel. The superimposed red and green channels are shown in the left column. In (**a**,**b**), the vesicle membrane exhibits an apparent membrane kink that reflects the limited spatial resolution of the optical microscope [11]. (With permission from ACS).

**Figure 4 membranes-13-00223-f004:**
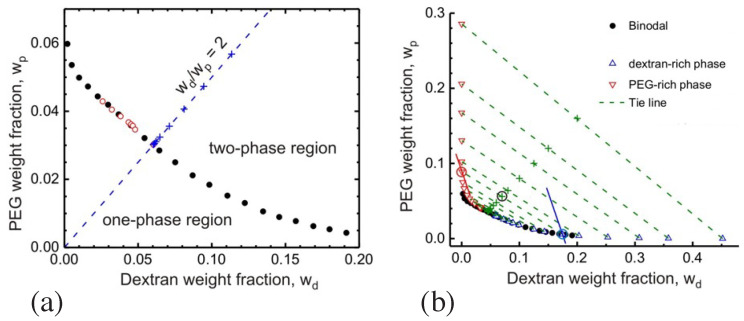
Phase diagram for aqueous PEG-dextran solutions at room temperature in terms of the weight fractions $w_{d}$ and $w_{p}$ of dextran and PEG. The binodal line (black and red data points) separates the one-phase region at low weight fractions from the two-phase region at higher weight fractions. The dashed line in (**a**) corresponds to constant weight fraction ratio $w_{d}/w_{p}=2$. The green dashed lines in (**b**) represent tie lines in the two-phase region. Each tie line has two end points which lie on the binodal. When the weight fractions are located on a certain tie line, the solution phase separates into a PEG-rich and a dextran-rich phase. The compositions of these two coexisting phases are given by the end points of the tie line as indicated by upward-pointing triangles for the dextran-rich phase and by downward-pointing triangles for the PEG-rich phase. These compositions can be determined from the measured mass densities of the two coexisting phases by constructing isopycnic lines of constant mass densities in the ($w_{d},w_{p}$)-plane. The intersections of these isopycnic lines with the binodal provide the comparisons of the coexisting phases. The blue and the red line segments represent isopycnic lines corresponding to the crossed data points (⊕) [16]. (With permission from ACS).

**Figure 5 membranes-13-00223-f005:**
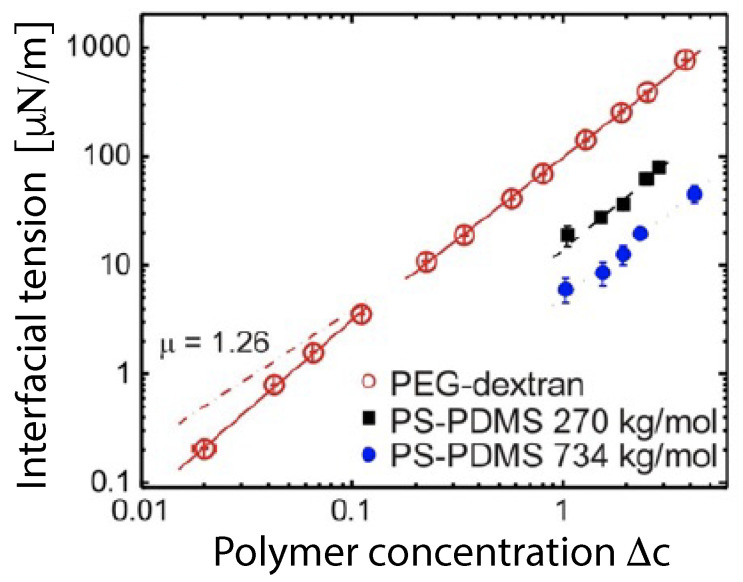
Interfacial tension $\Sigma_{\alpha \beta }$ of the liquid–liquid interface between the PEG-rich phase α and the dextran-rich phase β as a function of the polymer concentration $\Delta c\equiv \left(c-c_{\mathrm{cr}}\right)/c_{\mathrm{cr}}$ where $c_{\mathrm{cr}}$ denotes the concentration at the critical demixing point [16]. The red data for the PEG-dextran solutions exhibit the power-law behavior $\Sigma_{\alpha \beta }∼\Delta c^{\mu }$ where the critical exponent μ is close to the mean value μ=3/2. For comparison, the dashed red line corresponds to μ=1.26 based on the hyperscaling relation μ=2ν [49] where ν is the critical exponent of the correlation length. (With permission from ACS).

**Figure 6 membranes-13-00223-f006:**
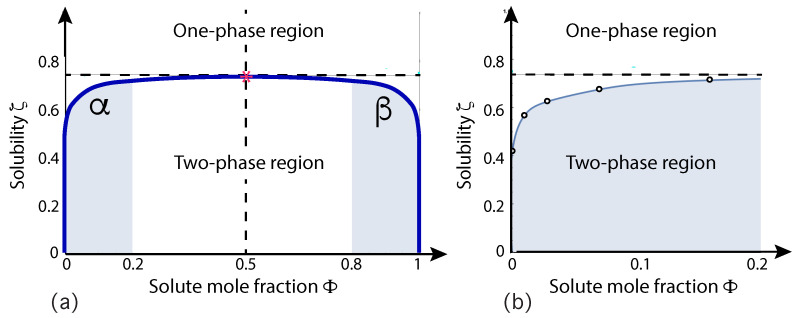
Phase diagram for a binary mixture of water and solute molecules as a function of solute mole fraction $\Phi_{S}$ and solubility ζ of the solute molecules in water [50]: (**a**) Global phase diagram for $0\leq \Phi_{S}\leq 1$. The phase diagram is mirror symmetric with respect to the dashed vertical line at $\Phi_{S}=1/2$, which implies horizontal tie lines. The critical demixing point (red star) with coordinates $\left(\Phi_{S},\zeta \right)=\left(1/2,0.746\right)$ is located at the crossing point of the dashed vertical line and the binodal line (dark blue), and (**b**) Phase diagram for $0\leq \Phi_{S}\leq 0.2$ corresponding to the grey-shaded region on the left of panel a. The four data points (open circles) on the binodal line have been determined by molecular dynamics simulations [50]. The binary mixture forms a uniform phase above the binodal line and undergoes phase separation into a water-rich phase α with $\Phi_{S}<0.5$ and a solute-rich phase β with $\Phi_{S}>0.5$. Essentially the same phase diagram is obtained when the solubility is replaced by the temperature.

**Figure 7 membranes-13-00223-f007:**
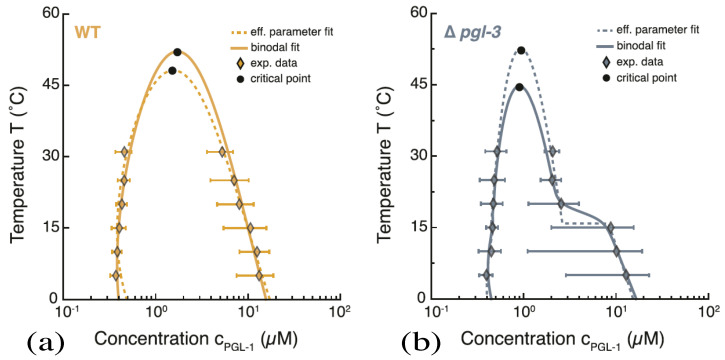
Phase diagram for condensates enriched in the protein PGL−1 labeled by GFP as observed in P granules of *C. elegans* cells [32]. The data in (**a**) were obtained for the wild type, those in (**b**) after the deletion of the protein PGL−3. The experimental data (diamonds) for the binodals are compared with theoretical binodals based on effective parameters for a binary liquid mixture, compare the phase diagram in Figure 6.

**Figure 8 membranes-13-00223-f008:**
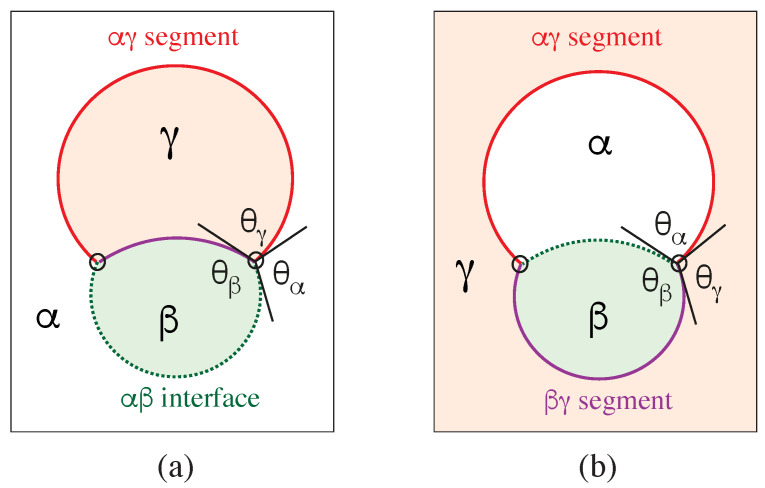
Apparent contact angles $\theta_{\alpha }$, $\theta_{\beta }$, and $\theta_{\gamma }$ for the vesicle-droplet systems in Figure 1: Phase separation of (**a**) exterior solution and (**b**) interior solution into the two liquid phases α and β. The contact angle $\theta_{\alpha }$ is the angle between the αβ interface (broken green line) and the αγ membrane segment (red line); the angle $\theta_{\beta }$ is the angle between the αβ interface and the βγ membrane segment (purple line); and $\theta_{\gamma }$ is the angle between the αγ and βγ membrane segments. The two contact lines, at which the three surface segments meet, are indicated by the four open circles in panel a and b.

**Figure 9 membranes-13-00223-f009:**
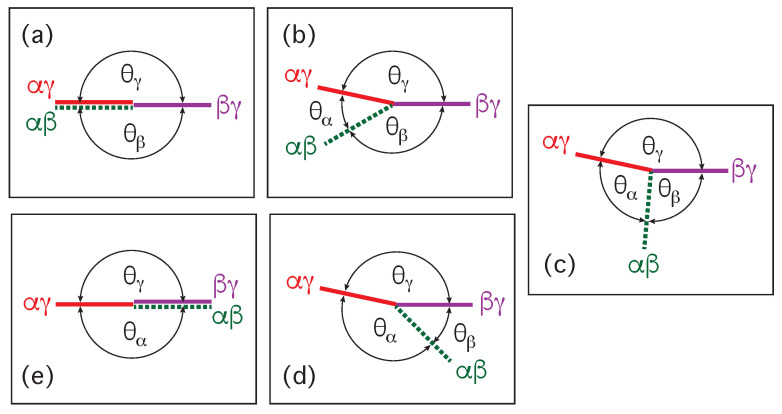
Apparent contact angles $\theta_{\alpha }$, $\theta_{\beta }$, and $\theta_{\gamma }$ for the adhesion morphologies in Figure 2: (**a**) Complete wetting by the α phase, which is equivalent to complete dewetting from the β phase, corresponds to the limit $\theta_{\alpha }=0$ and $\theta_{\beta }=\theta_{\gamma }=\pi$; (**b**) Partial dewetting from the β phase with $\theta_{\alpha }<\theta_{\beta }$; (**c**) Balanced adhesion with $\theta_{\beta }=\theta_{\alpha }$; (**d**) Partial wetting by the β phase with $\theta_{\beta }<\theta_{\alpha }$; and (**e**) Complete wetting by the β phase, which is equivalent to complete dewetting from the α phase, corresponds to the limit $\theta_{\beta }=0$ and $\theta_{\alpha }=\theta_{\gamma }=\pi$.

**Figure 10 membranes-13-00223-f010:**
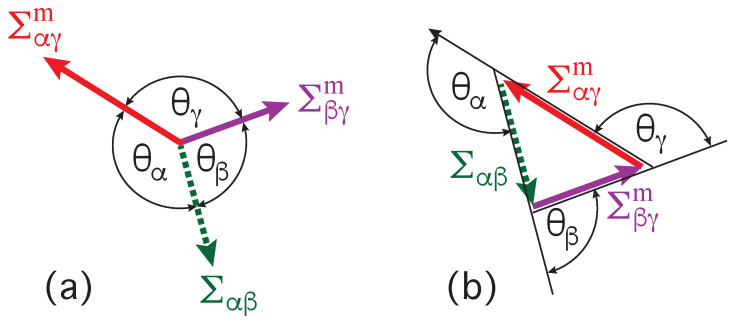
Force balance between the interfacial tension $\Sigma_{\alpha \beta }$ (green) as well as the two membrane segment tensions $\Sigma_{\beta \gamma }^{m}$ (purple), and $\Sigma_{\alpha \gamma }^{m}$ (red) for partial wetting by the β droplet which is characterized by the relationship $\theta_{\beta }<\theta_{\alpha }$ between the apparent contact angles $\theta_{\alpha }$ and $\theta_{\beta }$: (**a**) Each tension generates a force per unit length that pulls at the contact line in the direction of the corresponding arrow. The contact angles $\theta_{\alpha }$, $\theta_{\beta }$, and $\theta_{\gamma }$ have been introduced in Figure 8; and (**b**) In mechanical equilibrium, the three surface tensions must balance and form a triangle. The contact angles $\theta_{i}$ with i=α,β, and γ are the external angles of this triangle while the internal angles of the triangle are given by $\eta_{i}\equiv \pi -\theta_{i}$ [8].

**Figure 11 membranes-13-00223-f011:**
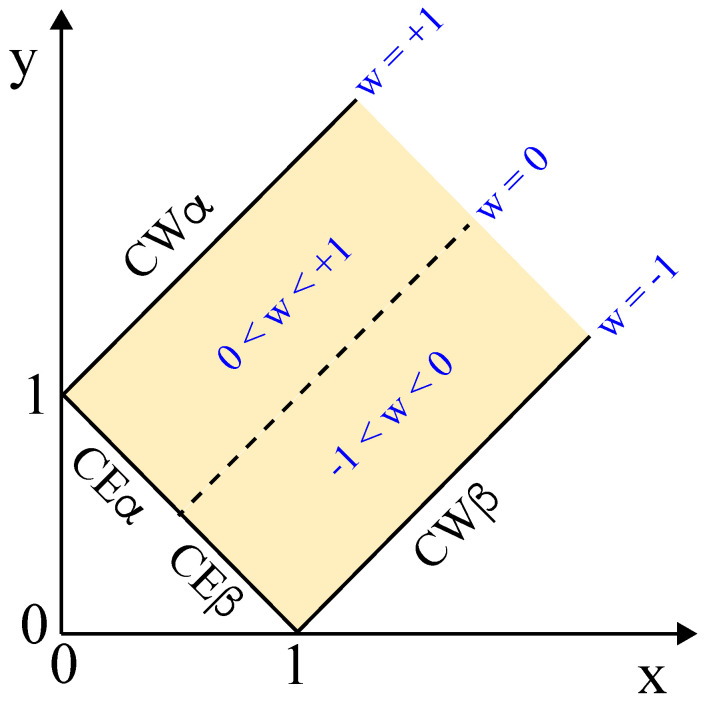
Force balance regime (yellow) and rescaled affinity contrast *w* as a tension ratios $x=\Sigma_{\alpha \gamma }^{m}/\Sigma_{\alpha \beta }$ and $y=\Sigma_{\beta \gamma }^{m}/\Sigma_{\alpha \beta }$, corresponding to the membrane segment tensions $\Sigma_{\alpha \gamma }^{m}$ and $\Sigma_{\beta \gamma }^{m}$ divided by the interfacial tension $\Sigma_{\alpha \beta }$. The rescaled affinity contrast *w* is defined in Equation (Equation 18). Within the yellow regime, the three surface tensions can balance each other along the contact line of droplet and vesicle. The force balance regime is bounded from below by the CWβ line of complete wetting of the vesicle membrane by the β phase with w=−1 and from above by the CWα line of complete wetting by the α phase with w=+1. The left boundary with y=1−x corresponds to complete engulfment of an α droplet (CEα) and to complete engulfment of a β droplet (CEβ), depending on the sign of the affinity contrast *w*. Balanced adhesion with w=0 (dashed line) divides the force balance regime up into a partial wetting regime by the β phase with −1<w<0 and a partial wetting regime by the α phase with 0<w<+1. The corner point with x=1 and y=0 corresponds to the limit of small segment tensions $\Sigma_{\beta \gamma }^{m}$, the corner point with x=0 and y=1 to the limit of small $\Sigma_{\alpha \gamma }^{m}$. Below the CWβ line, the vesicle avoids any contact with the α phase as in Figure 2e; above the CWα line, the vesicle has no contact with the β phase as in Figure 2a.

**Figure 12 membranes-13-00223-f012:**
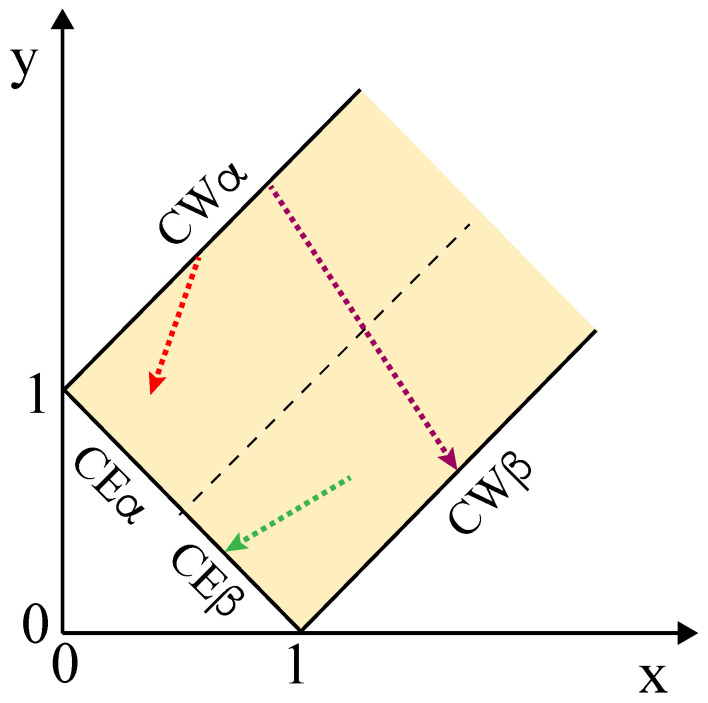
Morphological pathways of vesicle-droplet systems within the parameter space defined by the tension ratios $x=\Sigma_{\alpha \gamma }^{m}/\Sigma_{\alpha \beta }$ and $y=\Sigma_{\beta \gamma }^{m}/\Sigma_{\alpha \beta }$ as in Figure 11. The green pathway starts from partial wetting of the vesicle membrane by a β droplet and ends up with the complete engulfment of this droplet as in Figure 3c. The red pathway starts from complete wetting of the vesicle membrane by the α phase and then undergoes a complete-to-partial wetting transition, see the example in the next subsection. The purple pathway starts from complete wetting by the α phase and ends up with complete wetting by the β phase. For visual clarity, the different pathways have been drawn as straight lines but can, in general, be arbitrarily curved.

**Figure 13 membranes-13-00223-f013:**
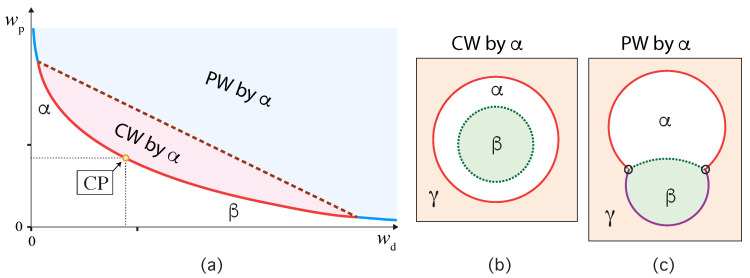
Schematic phase diagram and wetting morphologies of aqueous PEG-dextran solutions within giant vesicles, with the PEG-rich phase α and the dextran-rich phase β: (**a**) Phase diagram of PEG-dextran solutions at room temperature as in Figure 4. The phase diagram exhibits a one-phase region (white) at low weight fractions $w_{d}$ and $w_{p}$ of the two polymers and a two-phase region (light red and light blue) at higher weight fractions. The boundary between the one-phase and two-phase regions defines the binodal line which contains the critical demixing point (CP, orange). The two-phase region above the binodal is divided up into two subregions, a complete wetting (CW) subregion (light red) close to the critical point and a partial wetting (PW) subregion (light blue) further away from it. The boundary between the CW and PW subregions is provided by a certain tie line (purple dashed line), the precise location of which depends on the lipid composition of the membrane; (**b**) CW morphology and (**c**) PW morphology of the vesicle-droplet system corresponding to complete and partial wetting of the vesicle membrane by the α phase, corresponding to the light red and light blue subregions of the phase diagram in panel a [28].

**Figure 14 membranes-13-00223-f014:**
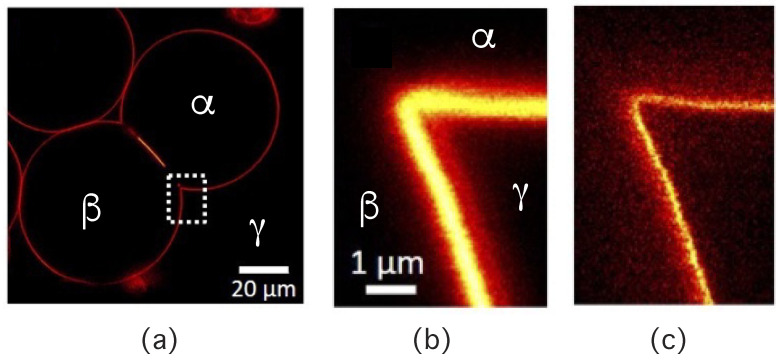
Apparent membrane kinks are low-resolution images of highly curved membrane segments: (**a**) GUV with an interior compartment that contains a PEG-rich α droplet and a dextran-rich β droplet. The region enframed by the white-dashed rectangle contains one membrane kink which is enlarged in panels b and c; (**b**) In the confocal microscope, the highly curved membrane segment cannot be resolved; and (**c**) In the STED image, the smoothly curved segment leads to a contour curvature radius of about 220 nm [52].

**Figure 15 membranes-13-00223-f015:**
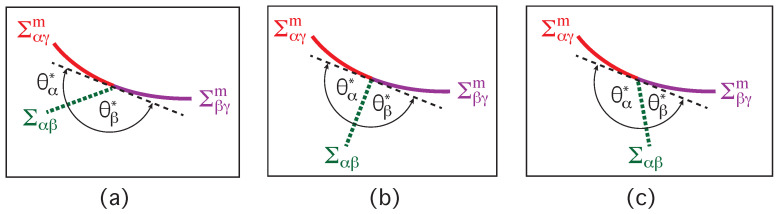
Intrinsic contact angles $\theta_{\alpha }^{*}$ and $\theta_{\beta }^{*}$ describing the force balance along the contact line for a smoothly curved membrane segment: (**a**) Partial dewetting of the β droplet with $\theta_{\alpha }^{*}<\theta_{\beta }^{*}$. The limit of zero $\theta_{\alpha }^{*}$ corresponds to complete dewetting from the β phase; (**b**) Balanced adhesion with $\theta_{\alpha }^{*}=\theta_{\beta }^{*}$; and (**c**) Partial wetting by the β droplet with $\theta_{\alpha }^{*}>\theta_{\beta }^{*}$. The limit of zero $\theta_{\beta }^{*}$ corresponds to complete wetting by the β phase. The dashed black line represents the common tangent plane of the two membrane segments at the contact line which implies $\theta_{\alpha }^{*}+\theta_{\beta }^{*}=\pi =180^{\circ }$. Same color code for surface segments and tensions as in Figure 9 and Figure 10.

**Figure 16 membranes-13-00223-f016:**
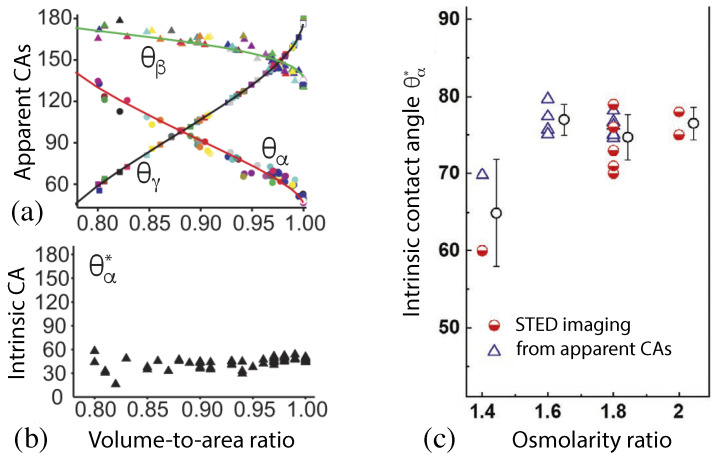
Experimental values of the intrinsic contact angle $\theta_{\alpha }^{*}$: (**a**) Apparent contact angles (CAs) for a batch of 63 GUVs with different volume-to-area ratios *v* as defined by Equation (Equation 48) [5]; (**b**) The intrinsic contact angle (CA) $\theta_{\alpha }^{*}$ as obtained from the apparent CAs in panel a by using Equation (Equation 47), which leads to $\cos\theta_{\alpha }^{*}=0.714\pm 0.075$ and $\theta_{\alpha }^{*}≃44.4^{\circ }$; and (**c**) Intrinsic contact angle $\theta_{\alpha }^{*}$ measured via STED imaging (half-filled circles) and compared to those calculated from the observed apparent contact angles via Equation (Equation 47) (open triangles) [52].

**Figure 17 membranes-13-00223-f017:**
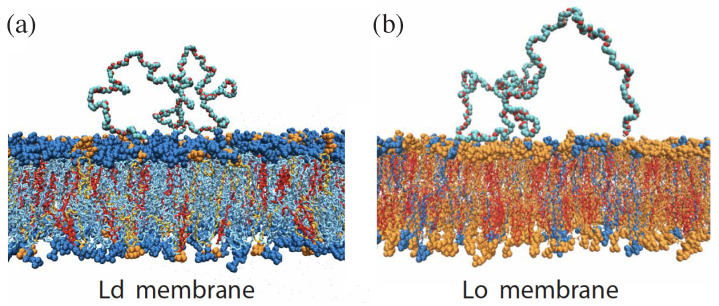
Typical conformations of a single PEG molecule adsorbed to two bilayers with different lipid compositions as observed in atomistic molecular dynamics simulations [9]. The color code for the lipids is blue for DOPC, orange for DPPC, and red for cholesterol. The lipid composition in (**a**) belongs to the liquid-disordered (Ld) phase, which is enriched in DOPC (blue), the one in (**b**) to the liquid-ordered (Lo) phase enriched in DPPC (orange). The PEG chains, which consist of 180 monomers, are only weakly bound to the lipid bilayers, with relatively short contact segments and relatively long loops in between two such segments. The two terminal OH groups of the PEG molecule are often bound to the membrane *via* hydrogen bonds. The same lipid compositions were studied experimentally in [9], but the polymer solution was semi-dilute and the PEG chains formed an adsorption layer close to the overlap concentration.

**Figure 18 membranes-13-00223-f018:**
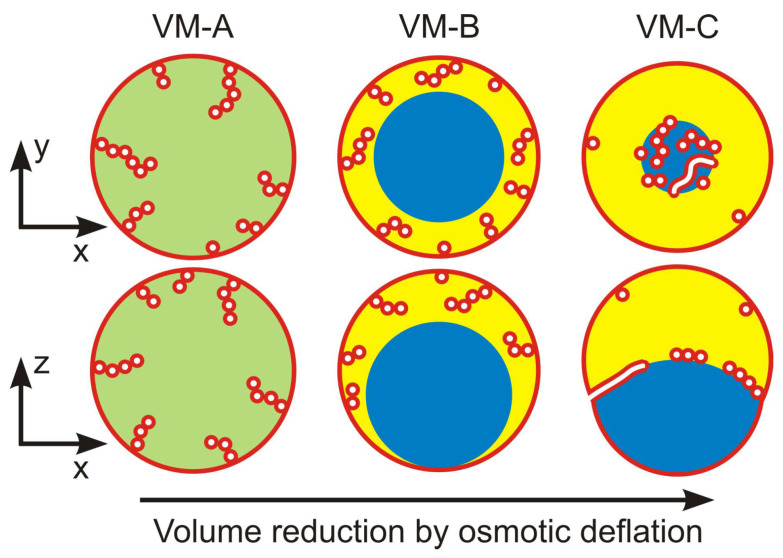
Three nanotube patterns corresponding to the distinct vesicle morphologies VM-A, VM-B, and VM-C observed along a deflation path that moves the interior PEG-dextran solution into the two-phase coexistence region: Schematic views of horizontal xy-scans (**top** row) and of vertical xz-scans (**bottom** row) across an individual vesicle, the volume of which is reduced by osmotic deflation. In all cases, the tubes are filled with the exterior solution (white). For the morphology VM-A, the interior polymer solution is uniform (green), whereas it is phase separated (blue-yellow) for the morphologies VM-B and VM-C, with complete and partial wetting of the membrane by the PEG-rich α phase (yellow). For the VM-B morphology, the nanotubes explore the whole PEG-rich α droplet but stay away from the dextran-rich β droplet (blue). For the VM-C morphology, the nanotubes adhere to the αβ interface between the two aqueous droplets, forming a thin and crowded layer at this interface [9].

**Figure 19 membranes-13-00223-f019:**
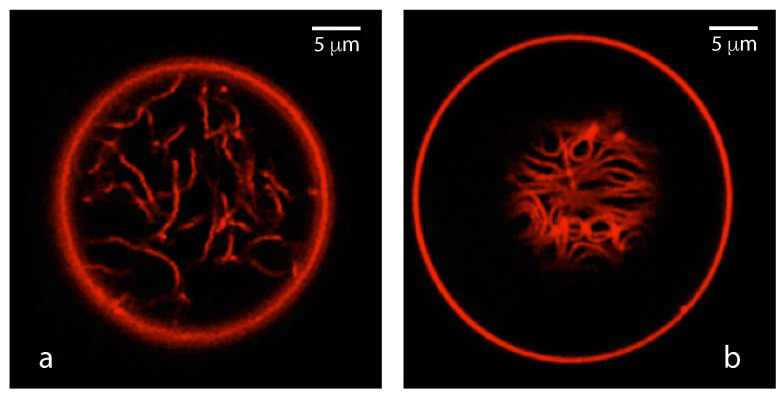
Patterns of flexible nanotubes formed by liquid-disordered membranes (red) exposed to aqueous solutions of PEG and dextran. All tubes protrude into the vesicle interior: (**a**) Disordered pattern of tubes freely suspended within the PEG-rich droplet enclosed by the vesicle, corresponding to the VM-B pattern in Figure 18; and (**b**) Thin layer of tubes adhering to the αβ interface between the PEG-rich and the dextran-rich phase, providing an example for the VM-C pattern in Figure 18. The width of the fluorescently labeled nanotubes is below the optical diffraction limit and of the order of 100 nm [9].

**Figure 20 membranes-13-00223-f020:**
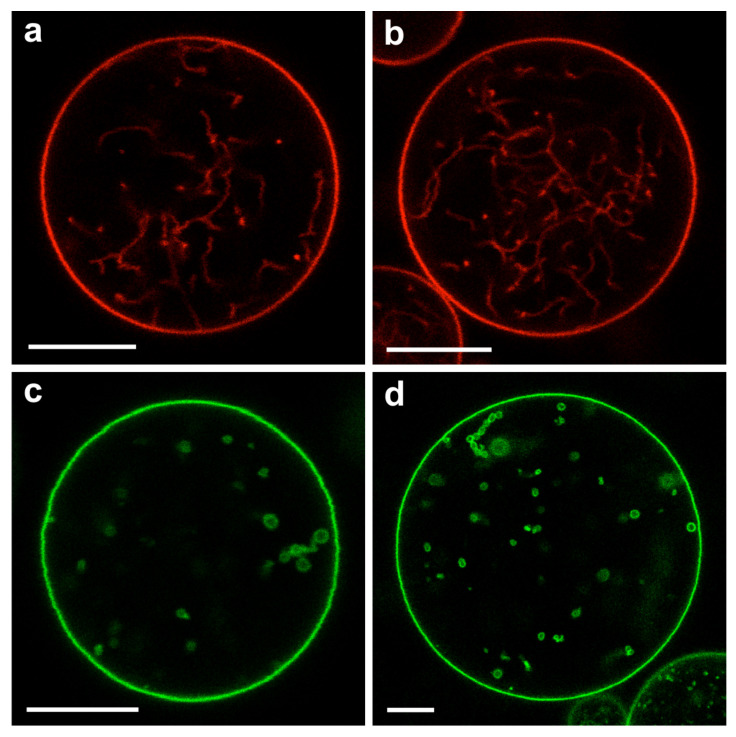
Nanotubes of GUV membranes with two different lipid compositions, which form a liquid-disordered lipid phase (red) in (**a**,**b**) and a liquid-ordered lipid phase (green) in (**c**,**d**). The two colors red and green arise from two different fluorescent dyes, which were added to the lipid bilayers using very small mole fractions. All vesicles are exposed to aqueous solutions of PEG 8000 and sucrose without dextran. The interior solution contains only PEG and no sucrose with the initial weight fraction $w_{p}=0.0443$ of PEG. The vesicles are deflated by exchanging the external medium by a hypertonic solution with no PEG but an increasing weight fraction $w_{\mathrm{su}}$ of sucrose. The vesicles in (**a**,**c**) are obtained for $w_{\mathrm{su}}=0.0066$, those in (**b**,**d**) for $w_{\mathrm{su}}=0.01$. The white scale bars are 10 μm in all panels [9].

**Figure 21 membranes-13-00223-f021:**
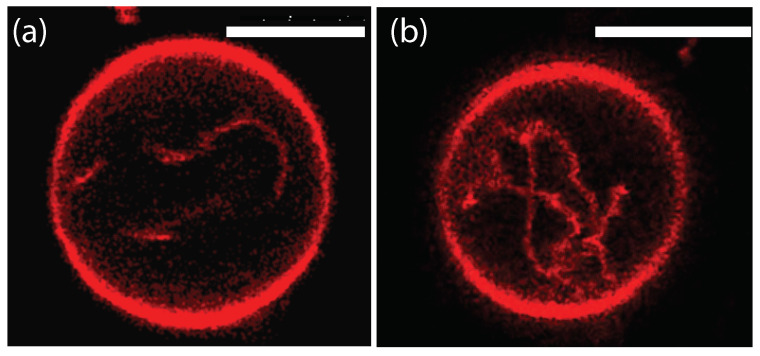
Membrane nanotubes protruding from the membranes of the mother vesicles (large red circles) into the vesicle interior in the absence of liquid–liquid phase separation. The vesicle membranes consist of the phospholipid POPC and the glycolipid GM1, with 2 mole % GM1 in (**a**) and 4 mole % GM1 in (**b**). The nanotubes are only visible when the membranes are doped with a fluorescently labeled lipid (red), in agreement with the theoretical analysis of micropipette experiments, which imply that the nanotubes have a width of the order of 100 nm. Scale bars: 10 μm [85].

**Figure 22 membranes-13-00223-f022:**
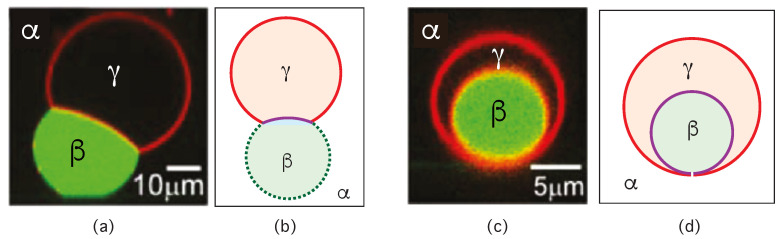
Microscopy images and schematic drawings for partial (**a**,**b**) and complete (**c**,**d**) engulfment of a condensate droplet β (green) by the membrane (red) of a giant vesicle [11]. For complete engulfment, the membrane forms two spherical segments that are connected by a narrow or closed membrane neck. This neck is not resolvable by conventional confocal microscopy but is indicated in the schematic drawing in (**d**). The color code in the drawings is the same as in Figure 2.

**Figure 23 membranes-13-00223-f023:**
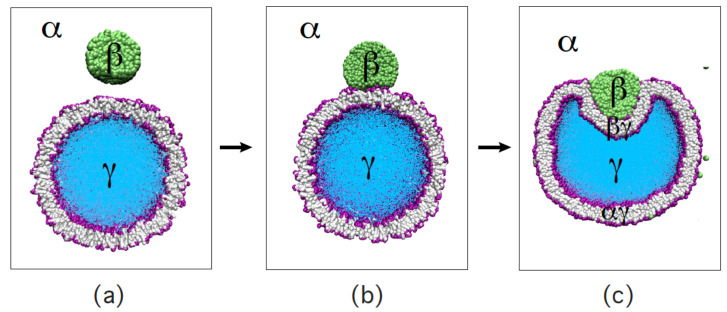
Partial engulfment of a condensate droplet (green) by the lipid bilayer (purple-grey) of a nanovesicle, as observed in molecular dynamics simulations [30]. The vesicle encloses the aqueous solution γ (blue). Both the nanodroplet and the nanovesicle are immersed in the aqueous bulk phase α (white): (**a**) Initially, the droplet is well separated from the vesicle which implies that the outer leaflet of the bilayer is only in contact with the α phase; (**b**) When the droplet is attracted towards the vesicle, it spreads onto the lipid bilayer, thereby forming an increasing contact area with the vesicle membrane; and (**c**) Partial engulfment of the droplet by the membrane after the vesicle-droplet couple has relaxed to a new stable state. The contact area between bilayer and β droplet defines the βγ segment of the bilayer membrane whereas the rest of the bilayer represents the αγ segment still in contact with the α phase. Vesicle and droplet have a diameter of 37 nm and 11.2 nm, respectively.

**Figure 24 membranes-13-00223-f024:**
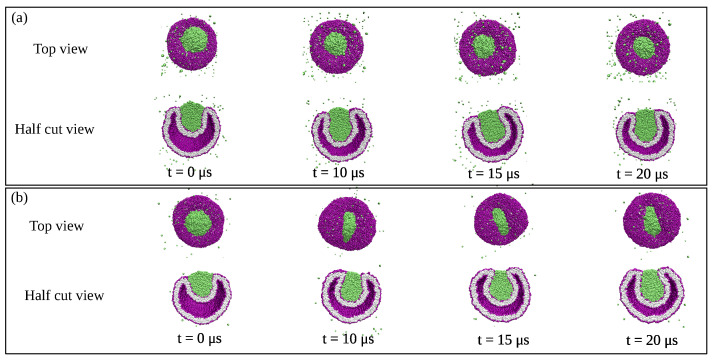
Stalled engulfment of large nanodroplets (green) by the vesicle membranes (purple-grey) as observed in molecular dynamics simulations [30]. Droplet engulfment can proceed in an axisymmetric or non-axisymmetric manner, depending on the lipid numbers, $N_{ol}$ and $N_{il}$, which are assembled in the outer and inner leaflets of the bilayer membranes: (**a**) For $N_{ol}=5400$ and $N_{il}=4700$, the engulfment process proceeds in an axisymmetric manner as can be seen from the circular shape of contact line and αβ interface (green); and (**b**) For $N_{ol}=5700$ and $N_{il}=4400$, both the contact line and the αβ interface attain a non-circular shape which implies a non-axisymmetric morphology of vesicle and droplet. The lipid numbers in (**b**) are obtained from those in (**a**) by reshuffling 300 lipids from the inner to the outer leaflet. Vesicle and droplet have a diameter of 37 nm and 19.6 nm, respectively.

**Figure 25 membranes-13-00223-f025:**
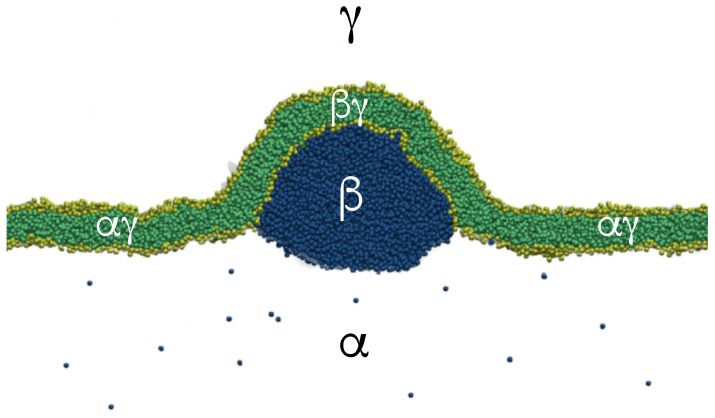
Partial engulfment of a condensate nanodroplet (β, dark blue) by a planar bilayer, consisting of lipids with yellow headgroups and green lipid tails as studied by molecular dynamics simulations [29]. The αβ interface between the droplet and the liquid bulk phase α forms a contact line with the bilayer which partitions this bilayer into a βγ segment in contact with the β droplet and into an αγ segment exposed to the α phase as in Figure 1a.

**Figure 26 membranes-13-00223-f026:**
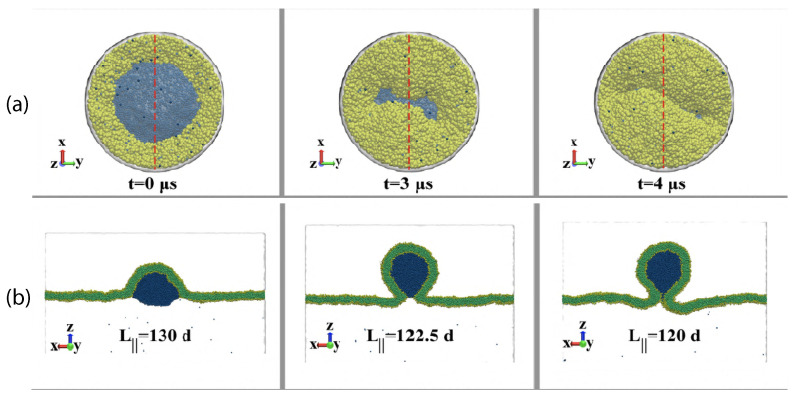
Formation of a non-circular, tight-lipped membrane neck generated by a nanodroplet (dark blue) that adheres to a planar bilayer [29]. This process was induced by a time-dependent reduction of the lateral size $L_{\|}$ of the simulation box, keeping the box volume fixed: (**a**) Bottom views of circular membrane segments (yellow) around the αβ interface (blue) of the β droplet, separated by the contact line which is circular at time t=0 μs, strongly non-circular after t=3 μs, and has closed into a tight-lipped shape after t=4 μs; and (**b**) Side views of the same membrane-droplet morphology, with perpendicular cross-sections through membrane (green) and droplet (blue) taken along the red dashed lines in panel (**a**). The non-circular shape of the membrane neck is caused by the negative line tension of the contact line and prevents membrane fission. The droplet has a diameter of about 12 nm. Same color code as in Figure 25.

**Figure 27 membranes-13-00223-f027:**
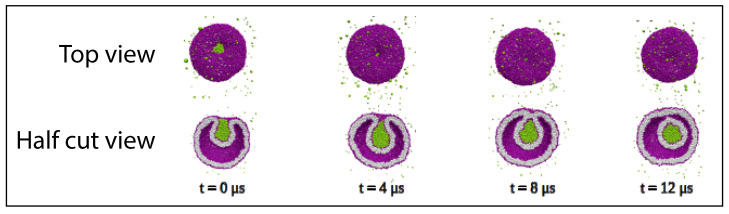
Endocytosis of condensate droplet (green) with complete engulfment of the droplet followed by division of the nanovesicle membrane (purple-grey) into two nested daughter vesicles as observed in molecular dynamics simulations [30]. In this example, the bilayer membrane consists of 5500 lipids in the outer and 4600 lipids in the inner leaflet. The contact line between membrane and droplet has a positive line tension $\lambda ≃+7\;k_{B}T/d$. The membrane neck closes at t=4 μs and undergoes fission at t=9 μs, generating a small intraluminal vesicle around the droplet. The undivided nanovesicle has a size of 37 nm, the droplet has a diameter of 11.2 nm.

**Table 1 membranes-13-00223-t001:** Relations between the apparent contact angles $\theta_{\alpha }$ and $\theta_{\beta }$ for the different wetting regimes displayed in Figure 9. The first column describes the wetting behavior of the α phase, the second column the wetting behavior of the β phase, and the last column the relation between $\theta_{\alpha }$ and $\theta_{\beta }$.

α Phase at Membrane	β Phase at Membrane	Contact Angles
complete wetting by α	complete dewetting from β	$\theta_{\alpha }=0$ and $\theta_{\beta }=\pi$
partial wetting by α	partial dewetting from β	$0<\theta_{\alpha }<\theta_{\beta }$
balanced adhesion		$\theta_{\alpha }=\theta_{\beta }$
partial dewetting from α	partial wetting by β	$0<\theta_{\beta }<\theta_{\alpha }$
complete dewetting from α	complete wetting by β	$\theta_{\beta }=0$ and $\theta_{\alpha }=\pi$

**Table 2 membranes-13-00223-t002:** Relationships between the three surface tensions for the different wetting regimes displayed in Figure 9. The first column describes the wetting behavior of the α phase, the second column the wetting behavior of the β phase, and the last column the corresponding relation between the surface tensions.

α Phase at Membrane	β Phase at Membrane	Surface Tensions
complete wetting by α	complete dewetting from β	$\Sigma_{\beta \gamma }^{m}=\Sigma_{\alpha \gamma }^{m}+\Sigma_{\alpha \beta }$
partial wetting by α	partial dewetting from β	$\Sigma_{\beta \gamma }^{m}<\Sigma_{\alpha \gamma }^{m}+\Sigma_{\alpha \beta }$
balanced adhesion		$\Sigma_{\alpha \gamma }^{m}=\Sigma_{\beta \gamma }^{m}$
partial dewetting from α	partial wetting by β	$\Sigma_{\alpha \gamma }^{m}<\Sigma_{\beta \gamma }^{m}+\Sigma_{\alpha \beta }$
complete dewetting from α	complete wetting by β	$\Sigma_{\alpha \gamma }^{m}=\Sigma_{\beta \gamma }^{m}+\Sigma_{\alpha \beta }$

## Data Availability

Not applicable.

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
