# Peer review of "Remodeling of Biomembranes and Vesicles by Adhesion of Condensate Droplets"

_membranes, 2023, doi:10.3390/membranes13020223_

Round 1

Reviewer 1 Report

The manuscript by Lipowsky et al. reviews the remodeling of interactions of membranes and condensate droplets. Interactions between phase-separated systems and biomembranes have gained attention as new findings indicated the diverse relation between condensate formation and membranes in living cells. A review of condensate droplets and biomembranes is needed to understand these relationships. This review specifically focuses on PEG/dextran phase-separated systems and their interaction with biomembranes. New findings in living cells are mentioned in a paragraph to connect the review with the current literature. I find it valuable to begin to review these crucial interactions, which provide means to control biological functions. This manuscript contributes to understanding contact angle, surface tension, and curvature, wetting. I recommend publication after the following comments have been addressed:

1. I am really having a hard time connecting all information presented with all types of condensate droplets. Phase separation is described in the paper but mostly focused on non-associative phase separation (i.e. PEG/dextran). There is also associative phase separation, which describes for instance complex coacervate formation. Complex coacervates are formed when you mix oppositely charged biomolecules. I am curious how the information presented will be relevant in the presence of charges. I would like to see a paragraph to comment on it. 

2. I see a paragraph that describes the recent developments including interactions between biomembrane and condensates in living cells. I would like to see in vitro studies that describe these types of interactions mentioned in this review as well. Papers that will fit in this description will include the followings:

https://onlinelibrary.wiley.com/doi/full/10.1002/ange.201914893

https://pubs.acs.org/doi/full/10.1021/acsnano.9b10167

https://pubs.acs.org/doi/abs/10.1021/jacs.0c12494

https://pubs.acs.org/doi/full/10.1021/jacs.2c04096

Author Response

Please find it in the attachment

Reviewer 2 Report

This review mentioned about the condensate droplets in contact with biomembranes and vesicles, and remodeling processes which are induced by the adhesion of condensate droplets arising from the phase separation in aqueous solutions. Recently, the phase separation in the cytoplasm has been studied frequently and its role, the mechanism of the phase separation, the behavior of phase separation etc have been clarified. Therefore, this review surely attracts the many researchers and attribute to the growth in this field. In the respect of the review paper, the content and survey about the recent papers are enough to summarize this topic. On the other hand, there are trivial revisions, I found.

(1)   Abstract: Line7, the author is only one person. So, "I" is better instead of “we”

(2)   Abstract line 14. Please delete the word of “194 < 200 words/”

(3)   Please write the definition of “affinity contrast” by equation.

(4)   About Fig 4(b), I found the undefined blue line. Please delete or mention about it.

(5)   Figure6, F“s” was disappeared.

(6)   Table 1. The bottom line was disappeared.

(7)   Figure 11, characters should be written as italic.

(8)   Figure 14, CA in y axis should be written as contact angle

Etc in other Figures.

So, I ask the author to carefully check his own manuscript.

These points are trivial but are better to be revised.

Author Response

Please find it in the attachment

Reviewer 3 Report

This review article by Reinhard Lipowsky describes the morphological changes in phospholipid bilayer membranes induced by the contact of a condensate droplet originating from liquid-liquid phase separation.  The subject of this review is very interesting and is also appropriate for the scope of the journal, Membranes.  Moreover, it is expected that this review will attract many researchers’ interest because of its relevance to a research subject of intense interest, i.e., molecular crowding.  So, I would like to recommend this review article for publication in this journal, but I think that this manuscript needs revising completely.

Overall,it seems to me that this review is basically based on a series of recent works by this author and colleagues (or collaborators), and I feel that this review is only a collective of those individual works that have been already published.  I mean that it is quite difficult for me to understand how relevant those individual works are to each other.  In addition, I think that this manuscript is too long, though this is a review article.  So, I would like to request the author to reorganize or restructure the entire manuscript in order to make it clear what is the point of this review as a whole.  For example, I think that it might be possible to merge Table 1 and Table 2 if the author restructure the manuscript.  What is described in Section 3.3 could be omitted because it has been already described in Introduction.  At least, I strongly request the author to describe clear motivation and justification of this review in Introduction and also to add more description about the author’s prospect for future research more sufficiently in the last section.

Author Response

Please find it in the attachment
